# Protein secondary structure in spider silk nanofibrils

Qijue Wang 📧 [1], Patrick McArdle[2], Stephanie L. Wang[2], Ryan L. Wilmington 📧 [2], Zhen Xing 📧 [2], Alexander Greenwood[1], Myriam L. Cotten[1], M. Mumtaz Qazilbash[2] & Hannes C. Schniepp 📧 [1✉]

Nanofibrils play a pivotal role in spider silk and are responsible for many of the impressive properties of this unique natural material. However, little is known about the internal structure of these protein fibrils. We carry out polarized Raman and polarized Fourier-transform infrared spectroscopies on native spider silk nanofibrils and determine the concentrations of six distinct protein secondary structures, including β-sheets, and two types of helical structures, for which we also determine orientation distributions. Our advancements in peak assignments are in full agreement with the published silk vibrational spectroscopy literature. We further corroborate our findings with X-ray diffraction and magic-angle spinning nuclear magnetic resonance experiments. Based on the latter and on polypeptide Raman spectra, we assess the role of key amino acids in different secondary structures. For the recluse spider we develop a highly detailed structural model, featuring seven levels of structural hierarchy. The approaches we develop are directly applicable to other proteinaceous materials.

[1] Department of Applied Science, William & Mary, P.O. Box 8795 Williamsburg, VA 23187-8795, USA. [2] Department of Physics, William & Mary, P.O. Box 8795 Williamsburg, VA 23187-8795, USA. ✉email: schniepp@wm.edu

Spider silk combines high strength and large extensibility with low density and is one of the most appealing materials found in nature[1,2]. Nanofibrils have long been suspected to play a pivotal role within the intricate, hierarchical silk structure of the protein fibers[1,3–23]. Surprisingly little, however, is still known about these silk nanofibrils, despite increased recent attention. Although silk nanofibrils have been observed in various silks, the structural model of silk fibers is still under development[7], with most models suggesting the presence of non-fibrillar phases and constituents, including an amorphous matrix phase[5,11], a shell/skin layer[5,11,17], as well as glycoprotein and lipid coatings[17,21]. Importantly, the prevalence of nanofibrils in silk fibers has not yet been generally established[7]. Recently, we demonstrated[6] that the strong and tough ribbon silk (Fig. 1a) of the recluse spider (Fig. 1b)[8,24] entirely consists of nanofibrils parallel to the ribbon's main axis (Fig. 1c and inset)[6]. The absence of any other structural elements in this silk showed that the mechanical properties of spider silk are already implemented at the level of an individual nanofibril[6]. Consequently, revealing the nanofibrils' structural makeup is arguably the most interesting question for a deeper mechanical understanding of silk. No other way of preparing native silk nanofibrils in pure form has been reported[7], and thus, the silk ribbons of the recluse spider provide a unique way to study nanofibrils. Protein secondary structure is probably the aspect of spider silk structure most challenging to determine. Experiments studying entire spider silk fibers have suggested that they are a semi-crystalline material, where β-sheet crystals are embedded in an "amorphous" matrix of helices, β-turns, and random coils[1,2,5,10,25,26]. However, since experiments with samples entirely and only consisting of nanofibrils have not been carried out before, the literature provides no concrete information about their secondary structures. Scattering experiments with entire fibers using X-ray, neutron, or electron sources have been used to estimate that β-crystals are only a few nanometers small[9,27,28], and to study their concentrations and crystal structures[27,29–31]. However, these scattering techniques have offered only very limited information about the non-crystalline structures, and they have not been used specifically on purely nanofibrillar samples. AFM or electron microscopy has not been able to detect secondary structures[32], due to resolution or contrast issues[21]. Instead, researchers have resorted to nuclear magnetic resonance (NMR)[33–35] and vibrational spectroscopy techniques[25,36–39].

Previous MAS NMR studies of $^{13}$C-labeled spider silk[33] from *Nephila clavipes* have suggested that the Gly-Gly-Ala motif forms disordered $3_1$-helices while the poly-Ala and poly-(Gly-Ala) regions adopt an ordered β-sheet structure. Similar studies[34] on dragline silk from *Argiope aurantia* specifically $^{13}$C/$^{15}$N-labeled on proline residues indicated that the Gly-Pro-Gly-X-X motif, found only in major ampullate spidroin 2 (MaSp2), adopts an elastin-like β-turn structure. Subsequently, further studies[35] found additional evidence for these structures as well as evidence for some α-helical structures in the poly-Gln-Gln-Ala-Tyr regions in the dragline silk of *Latrodectus hesperus* (black widow). These conclusions were all drawn from chemical shift information obtained from MAS NMR, either cross-polarization (CP) 1D[40] or 2D $^{13}$C-$^{13}$C DARR[41] or INADEQUATE[42] spectra or in the case of $^{13}$C/$^{15}$N-labeled proline, $^{15}$N/$^{13}$C HETCOR spectra[43].

Protein vibrational spectroscopy focuses on certain oscillation modes in the protein backbone. Raman spectroscopy, carried out with monochromatic visible light, probes the polarizability of vibrations, while FTIR is sensitive to their associated dipole moments. For materials with inversion symmetry, Raman and IR modes are mutually exclusive, making the two techniques complementary and sensitive to the specimen's symmetry[44–46]. Polarized Raman (p-Raman)[36,47] and polarized FTIR (p-FTIR)[38,48–50] have been separately applied to investigate the secondary structures of natural cylindrical spider silks and silkworm silks. Both the qualification and quantification of various structural elements have been examined. Similarly, with its high signal-to-noise ratio, synchrotron FTIR (S-FTIR) was also implemented to study the protein secondary structure in natural silk fibers[37,51]. However, due to the complex structural makeup of cylindrical spider silks and quasi-cylindrical silkworm silks, no direct spectroscopic information has been available for pristine silk nanofibrils.

The commonly studied vibrations are particularly sensitive to the local environment of the oscillator, and thus protein secondary structure[52–55]. This leads to characteristic peak shifts, which can be used to infer the secondary structure. This has, for instance, been observed through characteristic changes in the spectra of the simple polypeptides polyalanine, $(Ala)_n$, and polyglycine, $(Gly)_n$[52,56,57], when these polypeptides were prepared to feature different secondary structures. Silk proteins, in which Ala and Gly play important roles[58], have relatively rich sequences, and feature a mixture of different secondary

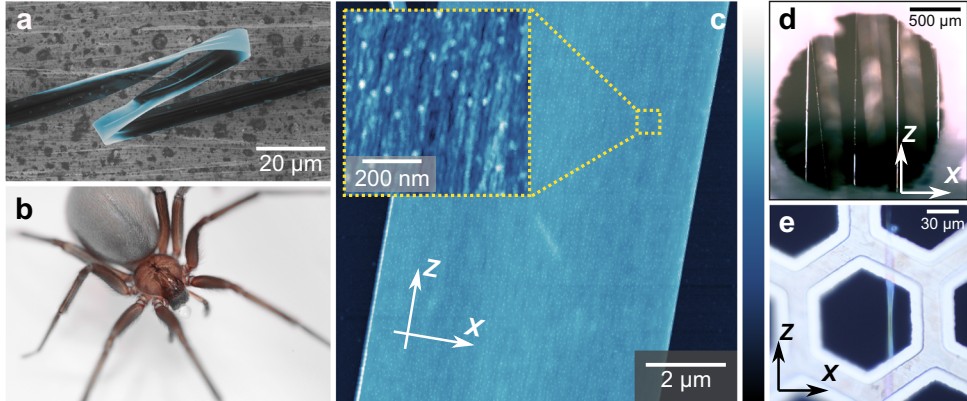

**Fig. 1 Loxosceles spider, its silk, and the investigated samples. a** False-colored scanning electron microscope (SEM) image of a strand of the ribbon silk of a *Loxosceles* (recluse) spider, which is entirely composed of nanofibrils ($n = 5$ independent SEM experiments)[6]. **b** A female Chilean recluse (*Loxosceles laeta*) spider. **c** Atomic force microscope (AFM) topography image of a *Loxosceles* silk ribbon ($n = 9$ independent AFM experiments). The white arrows indicate the "X" and "Z" directions in the plane of the ribbon, perpendicular and parallel to the ribbon axis, respectively. Inset: nanofibrils parallel to the ribbon axis Z are revealed at higher AFM magnifications. Color bars: 0–100 nm; inset: 0–12 nm. **d**, **e** Optical microscopy (OM) images of the FTIR (**d**) ($n = 6$ independent OM experiments) and Raman (**e**) ($n = 4$ independent OM experiments) samples. **e** Optical microscopy image of a location tested on the Raman sample.

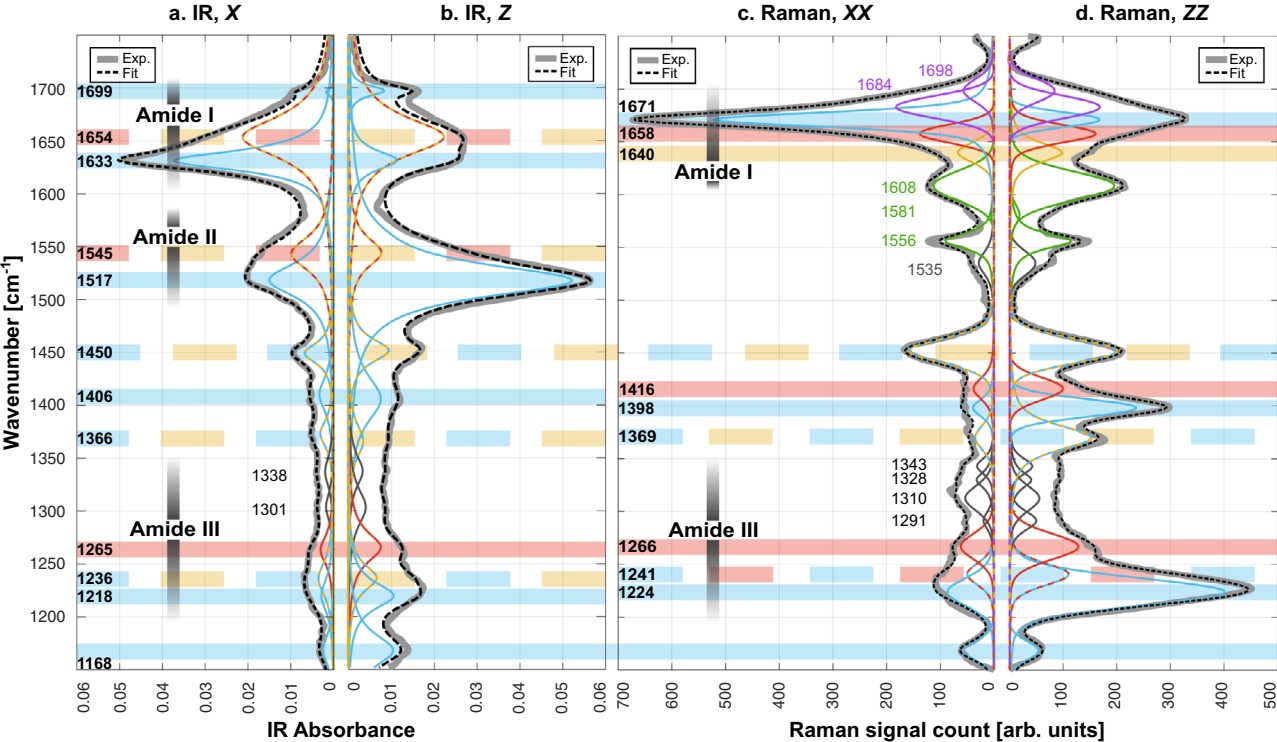

**Fig. 2 Polarized vibrational spectra.** p-FTIR (**a**, **b**) and p-Raman (**c**, **d**) spectra measured experimentally (Exp.) of single *Loxosceles* spider silk ribbons (thick gray lines) with polarizations in X and Z directions, in the plane of the ribbon. Dashed, black line: multi-peak fit; all constituting sub-peaks shown in colors. Blue, purple, red, and yellow peaks are assigned to β-sheet, β-turn, helical, and random coil peaks, repsectively. Green peaks represent amino acid residues with aromatic side-chain groups. Gray peaks are unassigned. The colored horizontal bars represent known peaks from the literature, with colors matching the secondary structure assignment.

structures. Consequently, their vibrational spectra have relatively broad, complex bands. A commonly used approach is to carry out peak deconvolution of these bands, in particular the amide-I/II/III bands in the 1140–1720 cm$^{-1}$ range[25,36–38,47,50,55,59,60]. Positions and heights of the obtained sub-peaks are compared to previous literature reports to determine the presence or prevalence of different secondary structures. There are, however, several significant problems with this approach leading to substantial ambiguity: first, the peak positions depend on the protein sequence, which complicates matching with results carried out on different proteins[52,61]; second, there may be interference with unrelated peaks in the same spectral region[52,61]; and finally, there are conflicting interpretations of certain peaks in the literature[25,36–38].

Here, we overcome these limitations with a combination of p-Raman and p-FTIR spectroscopy on the same type of silk to gain a much greater wealth of vibrational data than in previous studies of silk. Exploiting the opportunity to collect vibrational spectra exclusively from native silk nanofibrils and using fully oriented samples allows us not only to reveal the composition of their secondary structures but also to determine these secondary structures' 3-dimensional angular orientations within the nanofibrils. We show that β-sheets are the dominating secondary structure, but helical structures also play a significant role, with a simultaneous presence of 3$_1$-helices and α-helices in spider silk. Our findings are fully supported by magic-angle spinning nuclear magnetic resonance spectroscopy (MAS NMR). Furthermore, we consider a superposition of spectra from (Ala)$_n$ and (Gly)$_n$ polypeptides, to approximate our measured silk spectra, in order to assess the contribution of specific amino acids to the observed silk spectra and secondary structures.

## Results and discussion

We acquired p-FTIR and p-Raman spectra by orienting the incident beam normal to individual spider silk ribbons, such that its polarization was aligned with the X or Z axes of the ribbon (see Fig. 1 and Supplementary Figure 1). The obtained IR$_X$ and IR$_Z$ spectra are featured in Fig. 2a/b (gray, thick lines). For the p-Raman measurements, and additional analyzer was placed in the detection path, initially oriented parallel to the excitation. The corresponding Raman$_{XX}$ and Raman$_{ZZ}$ spectra are shown in Fig. 2c/d. ("Cross" polarized spectra will later be called Raman$_{XZ/ZX}$, Supplementary Figure 2.) The p-Raman and p-FTIR spectra show a significant dependence on polarization, demonstrating strong anisotropy of the silk samples. The goal is to understand these spectra as completely as possible by expressing them as a superposition of the proteins' vibration modes, each mode resulting in a peak of a well-defined shape. Multi-peak fits to the experimental spectra (dashed black lines in Fig. 2a–d, see details in Methods) approximate the experimental data well; all constituting sub-peaks are shown in thinner lines of different colors in all panels. We will link the following six pieces of information for a coherent interpretation of this data that is in full agreement with the available information in the literature: polarization dependence of individual sub-peaks from our p-Raman and p-FTIR spectra, structural and orientational information of our sample, geometric and orientational information on specific protein vibrations, known peak positions of polarized and unpolarized protein spectra in the literature, measured and calculated spectra of simple polypeptides consisting of the amino acids dominating in spider silk, and MAS NMR spectra of the same silk.

Figure 2 shows the most important spectral region, 1150–1750 cm$^{-1}$, with the amide-I/II/III bands indicated, each

representing well-defined vibrations of the protein backbone (wider-range spectra provided in Supplementary Figure 3)[52–54,61]. For an overview, let us first consider the amide-I band (≈1610–1700 cm$^{-1}$), which is most commonly used to determine protein secondary structures[38,47]. It is mainly due to a simple stretching vibration of the C=O bond in the protein backbone (see Fig. 3a)[52,54,61]. In agreement with the literature[38,50], the IR$_{X/Z}$ amide-I peaks feature two sub-peaks generally attributed to β-sheets (1633 cm$^{-1}$ and 1699 cm$^{-1}$, indicated in light blue) and one sub-peak at 1654 cm$^{-1}$. In the literature, the latter peak is attributed to "amorphous" structures, which does not only include truly random coil structures, but also helices. Apparently, the two types of secondary structure do not give rise to distinct peaks; in accordance with our coloring scheme used below, this peak is shown in yellow/red dash. The Raman$_{XX/ZZ}$ amide-I band needs 5 sub-peaks[25,36,47], generally attributed to random coil (yellow, 1640 cm$^{-1}$), helical (red, 1658 cm$^{-1}$), and β-sheet (blue, 1671 cm$^{-1}$) structures, in agreement with the literature. The understanding of the Raman sub-peaks at 1684 cm$^{-1}$ and 1698 cm$^{-1}$ (violet) is less established[47]; they have been suggested to represent type-III and type-I β-turns, respectively[25,47,59]. Most notably, this initial analysis of our spectra in the amide-I range already establishes that the spider silk nanofibrils contain β-sheets and helices. Moving to lower frequencies, the IR$_{X/Z}$ amide-II band (≈1500–1570 cm$^{-1}$) can be fitted well by two peaks at 1517 cm$^{-1}$ and 1545 cm$^{-1}$, colored blue (β-sheet) and yellow–red (amorphous) in Fig. 2a/b, respectively. Interestingly, amide-II IR peaks of silk have only rarely been discussed in the literature[49,50]. The commonly accepted vibration mode associated with the amide-II peak is shown in Fig. 3a/b/d, essentially a stretch of the bond between the carbonyl carbon (gray) and the nitrogen (green) atoms of the protein backbone, also causing a lateral motion of the hydrogen atom (blue) attached to the N[52,54,61]. While the amide-II band is rarely observed in Raman[52,61], our Raman$_{XX/ZZ}$ spectra feature three peaks in a similar spectral region (1556 cm$^{-1}$, 1581 cm$^{-1}$, and 1608 cm$^{-1}$, in green). However, they are due to aromatic side chains[47,52,62] and will not be further considered here (see Supplementary Section 4). The amide-II vibrational modes are only IR active and therefore odd parity (antisymmetric under inversion). The amide-III band is by far the most complicated one and has only rarely been used for structure determination via vibrational spectroscopy of spider silk[37]. The associated spectral band (≈1250–1350 cm$^{-1}$) is feature-rich, so only few have attempted peak deconvolution and assignment, while others have considered it hopeless[54]. As shown in Fig. 3, the vibrational modes giving rise to this frequency band vary significantly for the different secondary structures. In β-sheets (Fig. 3a) and α-helices (Fig. 3d), it involves an N–H in-plane bending, involving neighboring C atoms. For the (Gly)$_n$-II 3$_1$-helix (Fig. 3b), the two C$_\alpha$–H-bonds make an in-phase ("wag") or out-of-phase ("twist") bend normal to the H–C$_\alpha$–H plane. Our amide-III deconvolution is based on 6 peaks for both IR and Raman spectra. We use a consistent coloring scheme for sub-peaks — blue: β-sheets, red: helices, violet: β-turns, yellow: random coil, green: side chains, gray: unassigned. For peaks showing the same position for two secondary structures, we use dashed lines with alternating colors of the two structures.

The analysis presented so far is in line with the state of the art in vibrational spectroscopy of silk. However, using the polarization information, we can perform a significantly deeper analysis. The way our IR data is displayed in Fig. 2, the IR$_X$ and IR$_Z$ pair would exhibit mirror symmetry with respect to the vertical axis in the absence of polarization anisotropy. The same applies to the Raman$_{XX/ZZ}$ pair. Our spectra, however, are evidently highly anisotropic. For instance, the 1633 cm$^{-1}$ β-sheet component of the IR amide-I band is much more pronounced in the $X$ direction than in the $Z$ direction. Hence, the dipoles of the C=O bonds

associated with this peak feature an average orientation preference in $X$ over $Z$; the 1671 cm$^{-1}$ amide-I β-sheet component of the p-Raman data shows a corresponding orientation preference. To obtain more information about the 3-dimensional orientation of the protein structures, we also collected Raman$_{YY/ZZ}$ spectra on ribbons rotated axially by 90°, such that the laser polarizations were parallel to the ribbons' $Y$ or $Z$ directions (see Supplementary Figure 2). Notably, the Raman$_{YY}$ spectra were very similar to Raman$_{XX}$, indicating that the protein secondary structures essentially feature local rotational symmetry around the $Z$ axis — the axis of both the ribbon-shaped fiber and the constituting nanofibrils. Consequently, the C=O bonds are primarily oriented normally to the fiber axis based on amide-I analysis of Raman$_{XX/YY/ZZ}$ spectra. In contrast, the 1640 cm$^{-1}$ p-Raman amide-I peak, generally associated with random coil structures, featured similar strength in $X$, $Y$, and $Z$ polarizations. Any other finding would have been surprising, since a random coil organization cannot give rise to polarization preference. For the 1671 cm$^{-1}$ amide-I β-sheet peak, in contrast, the observation of polarization anisotropy is sensible, since β-sheet crystals are highly organized structures, where bonds have well-defined orientations; C=O bonds are parallel, at a right angle with the axis of the protein backbone, which is also defined as the axis of the β-sheet. Correspondingly, our spectra show that the β-sheets are oriented predominantly parallel to the fiber axis. Hence, our orientational analysis of the amide-I data confirms the commonly used assignments and is fully consistent. As shown in the following, applying our method across all measured spectra has allowed us to make structural assignments not known in the previous literature and clarify some controversially discussed cases. For quantitative analysis of the $X/Z$ polarization anisotropy, we calculated $P_{ZX} = \frac{|Z|-|X|}{|Z|+|X|}$ for each peak, where $|X|$ and $|Z|$ are the peak heights in the two polarizations. This metric has the advantage over a previous approach[50] that it is symmetric in $X$ and $Z$, and $|P_{ZX}|$ is the degree of orientation. It is $P_{ZX} = 1$ for perfect alignment in $Z$ direction, $P_{ZX} = -1$ for perfect alignment in $X$ direction, $P_{ZX} = 0$ when there is no polarization preference in $X$ or $Z$ directions, and $P_{ZX} \in (-1,1)$ for partial polarization. The results in Fig. 4a represent $P_{ZX}$ values for IR peaks with circles; values for Raman peaks are shown as crosses with error bars (see details in Supplementary Section 5).

Notably, both the 1671 cm$^{-1}$ Raman and 1633 cm$^{-1}$ IR peaks feature $P_{ZX}$ values of ≈ −0.6. This is fully consistent with our assignment that both peaks represent β-sheets vibrating in the same basic mode (amide-I). For β-sheets aligned perfectly with the fiber axis with all C=O bonds oriented fully perpendicular to the β-sheet axis, $P_{ZX} = -1$ would be expected. The most likely explanation for the lower degree in polarization anisotropy is that not all β-sheets are oriented perfectly with the $Z$ axis (see Fig. 4b). To estimate the degree of angular misalignment, we assumed that the angles of sheets with the $Z$ axis are distributed Gaussian (see Fig. 4c) and calculated the corresponding polarization anisotropy (details described in Supplementary Section 6). We found that $P_{ZX} = -0.6$ is obtained when the angular distribution is centered around the $Z$ axis (0°) with a spread of ±34.1°. This method generally allows us to characterize the degree of orientation of protein secondary structures via vibrational spectroscopy.

The amide-I IR peak at 1699 cm$^{-1}$ is less commonly discussed[48,50,55,60], but is usually also assigned to β-sheets; surprisingly, it features opposite orientation ($P_{ZX} \approx +0.6$). Essentially, the amide-I peak observed in Raman at 1671 cm$^{-1}$ appears as two peaks at 1633 cm$^{-1}$ and 1699 cm$^{-1}$ in the IR spectra due to transition dipole coupling (TDC) with C=O groups in adjacent peptide strands[52,58,63,64]. This leads to two variants of the same amide-I mode in IR with opposite polarizations (see

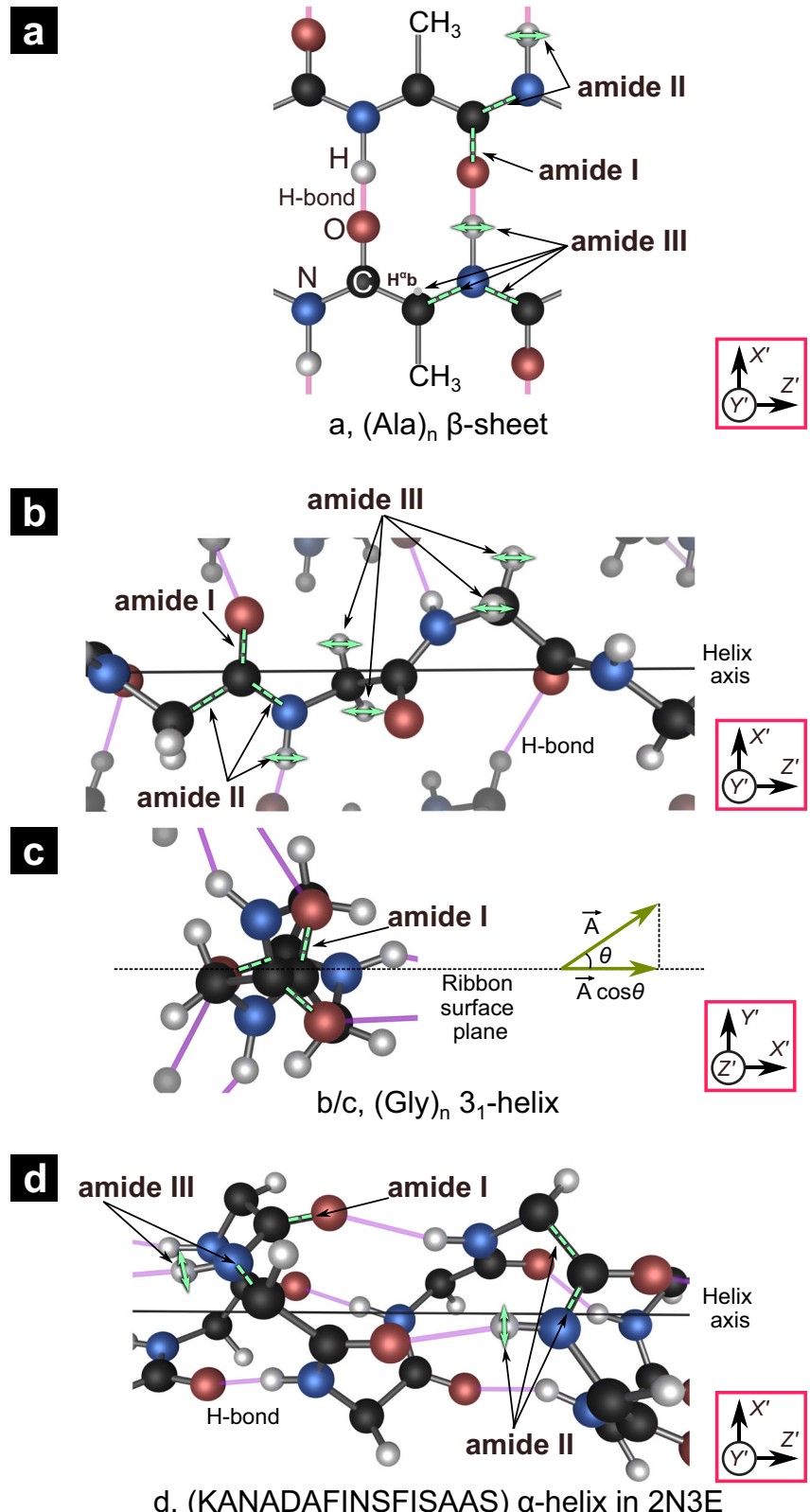

**Fig. 3 Vibrations associated with amide-I/II/III bands for different secondary structures.** Structure and amide-I/II/III vibrational modes in **a** an antiparallel β-(Ala)$_n$ β-sheet[52,58,63]. **b/c** a (Gly)$_n$-II 3$_1$-helix[82]. **b** Side view (neighboring strands dimmed; not all hydrogen bonds shown). Black solid line: helix axis. **c** Top view of a single 3$_1$-helix (black, dashed line: recluse ribbon surface). **d** α-helix segment (KANADAFINSFISAAS) in the protein 2N3E (side view)[83]. Black, blue, red, and white spheres represent carbon (C), nitrogen (N), oxygen (O), and hydrogen (H) atoms, respectively. H-bonds are represented by light purple rods. The α-hydrogens and functional groups are not shown for β-sheet and α-helix, respectively. Amide-I/II/III modes are indicated by green dashed lines and arrows. The axes in the red boxes represent the local orthogonal coordinates.

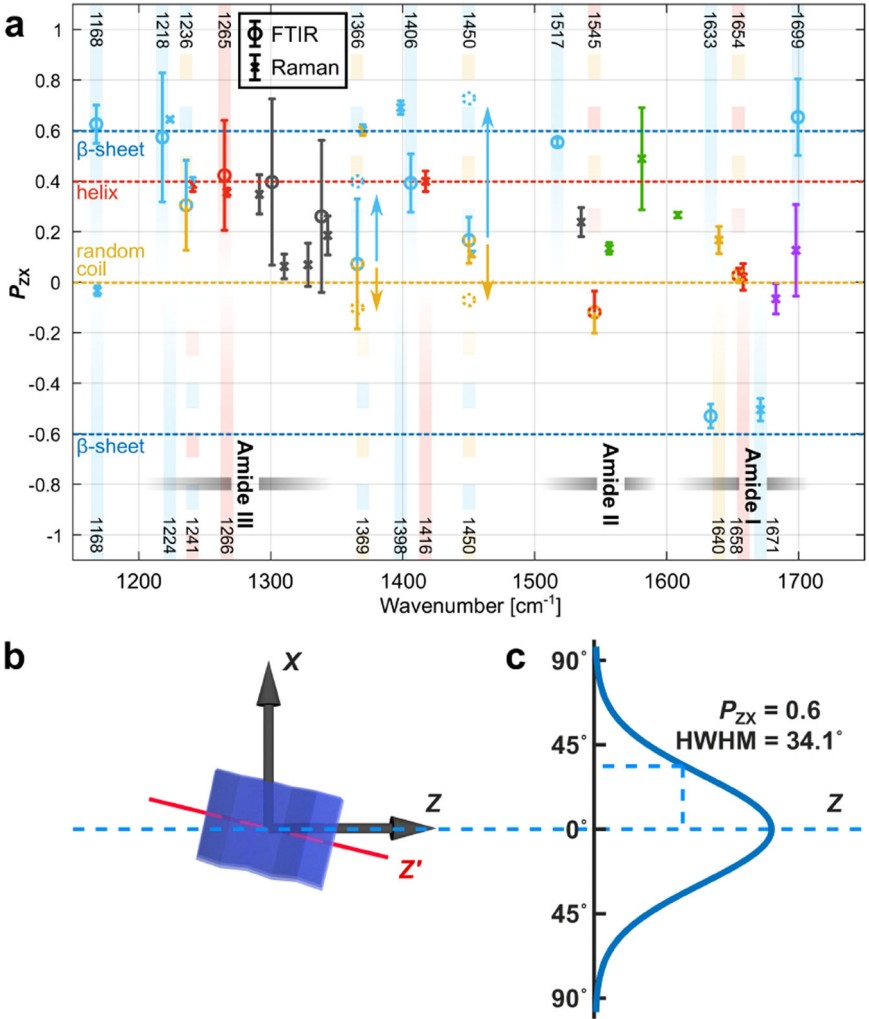

**Fig. 4 Orientational characteristics. a** $P_{ZX}$-values calculated for each FTIR (circles) and Raman (crosses) peak. Color assignments between each color on the peaks and corresponding protein structure are same as Fig. 2. Vertical gradient bands indicate the postion of FTIR (upper half) and Raman (lower half) peaks in corresponding colors, respectively. The horizontal yellow, red, and blue dashed lines are located at $P_{ZX}=0$, 0.4, and ±0.6, respectively. The IR and Raman data points are arithmetic mean values. Vertical error bars on IR data points are derived by minimizing mean squared error (MSE) and error propagating method. Vertical error bars on Raman data points are standard deviations of $P_{ZX}$ values calculated from repeated noisy fitting processes (number of fittings, $n_F = 15$) (see details in Supplementary Section 5). Treating β-sheet (≈44%) and amorphous (≈56%) fractions separately for the 1366 cm$^{-1}$ and 1450 cm$^{-1}$ peaks leads to split $P_{ZX}$ values, one for each fraction, indicated by arrows and dotted circles. **b** A β-sheet nanocrystal model featuring an angle between its internal symmetry axis Z' and the Z axis of the ribbon (the axis of the nanofibrils). **c** When $P_{ZX}=0.6$, the orientation distribution of β-sheets within the nanofibrils features a normal distribution with a half width at half maximum (HWHM) of 34.1°.

Supplementary Section 7)[52,63]. Importantly, $|P_{ZX}| \approx 0.6$ consistently for all three major amide-I β-sheet peaks in Raman and IR.

Orientational analysis of the helical peaks is complicated by the fact that two types of helices have been suggested to be present in spider silk[33,34,37,38,65,66]. NMR analysis has shown strong evidence for the presence of $3_1$-helices[33,34,65], while other studies have concluded a significant α-helical content[37,38]. Interestingly, the orientation of bonds in the two types of helices is so different (see Fig. 3b/d) that very different angular response patterns in vibrational spectroscopy can be expected. For $3_1$-helices, the C=O bonds chiefly responsible for the amide-I peak are oriented nearly perpendicular to the helix axis, from which they are pointing away approximately radially (Fig. 3c). For α-helices, in contrast, the C=O bonds make a significantly smaller angle with the helix axis, resulting in a significant dipole component parallel to the helix axis (Fig. 3d). We fully confirmed this qualitative assessment by calculating $P_{ZX}$ of the amide-I mode for α- and $3_1$-helices perfectly

oriented in Z direction using published coordinates of helical spider silk sequences, yielding +0.708 and −0.733, respectively (details in Supplementary Section 8). Indeed, the two helix types feature opposite polarization preference. Interestingly, both the helical amide-I Raman peak at 1658 cm$^{-1}$ and the amorphous amide-I IR peak at 1654 cm$^{-1}$ (featuring contributions from helices and random coil) show no significant orientation preference, $P_{ZX} \approx 0$, which might easily be interpreted to the effect that helices in spider nanofibrils are randomly oriented. However, since α- and $3_1$-helices show opposite polarization preference, an alternate scenario leading to $P_{ZX} \approx 0$ is a mixture of oriented α- and $3_1$-helices. Considering the amide-I peak alone, the two scenarios cannot be distinguished. A resolution can be found by considering the amide-III band featuring three vibrational peaks also commonly associated with helices—the IR peak at 1265 cm$^{-1}$ and the Raman peaks at 1266 cm$^{-1}$ and 1241 cm$^{-1}$—as well as another helical Raman peak at 1416 cm$^{-1}$. Interestingly, all four peaks uniformly feature $P_{ZX} \approx 0.4$, a significant polarization in Z direction. This

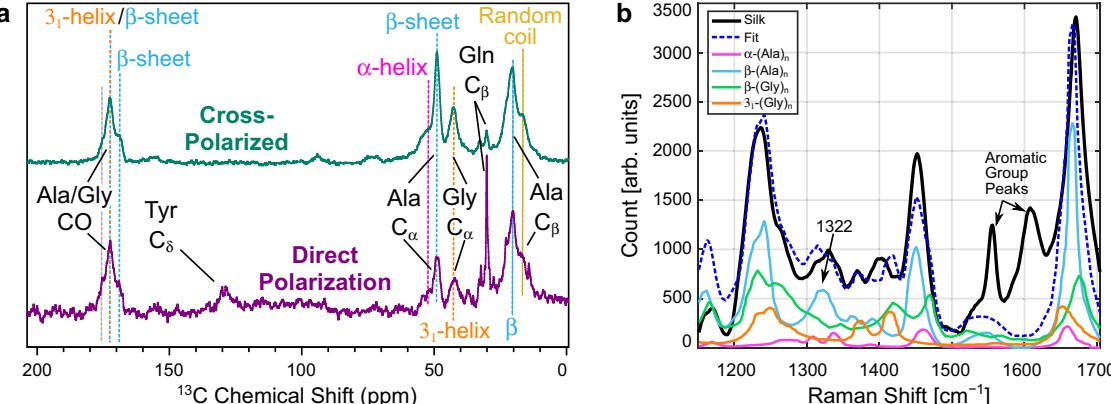

**Fig. 5 Silk nuclear magnetic resonance (NMR) results and polypeptide spectral superposition. a** Comparison of cross-polarized (teal) and direct-polarization (purple) [13]C MAS NMR spectra of *Loxosceles* dragline silk. Typical peak positions for Ala and Gly residues for different secondary structures in dashed lines, with blue, orange, pink, and yellow colors representing β-sheet, $3_1$-helix, α-helix, and random coil. Peak assignments based on comparisons to previously published data on silk are given in Table 1. **b** Comparison of pseudo-unpolarized Raman spectra from purely nanofibrillar spider silk (black line) and model polypeptides (colored solid lines[69,71], blue: β-(Ala)ₙ, green: β-(Gly)ₙ, orange: $3_1$-helix, pink: α-helix). The blue dashed line fits the silk spectrum with a linear combination of polypeptide spectra.

**Table 1 Comparison of observed [13]C shifts and known values for amino acids in a given secondary structure.**

| Amino acid | Peak | Recluse silk [ppm] | Literature assignment[a] | Range [ppm][a] |
|---|---|---|---|---|
| Ala | $C_\alpha$ main peak | 48.9 | β-sheet | 48.2–49.3 |
| Ala | $C_\alpha$ shoulder | 53.3 | α-helix | 52.3–52.8 |
| Ala | $C_\beta$ main peak | 20.4 | β-sheet | 19.9–20.7 |
| Ala | $C_\beta$ shoulder | 16.7 | Random-coil | 15.7–17.1 |
| Ala | CO[b] | 172.5 | β-sheet | 171.6–172.4 |
| Gly | $C_\alpha$ | 42.7 | $3_1$-helix | 41.4–42.5 |
| Gly | CO[b] | 172.5 | $3_1$-helix | 171.2–173.1 |
| Gly | CO shoulder | 169.5 | β-sheet | 168.4–169.7 |

[a]Obtained from refs. [33,66–68].
[b]Unresolved between Ala and Gly.

observation implies oriented helices, and thus, the $P_{ZX} \approx 0$ observation for the vibrational amide-I peaks can only be explained by a coexistence of both helix types, which has not previously been reported or suggested for spider silks.

We further calculated $P_{ZX}$ for the amide-III mode of α- and $3_1$-helices oriented in Z direction and found −0.528 and +0.894, respectively (see Supplementary Section 8). While our results did not allow us to determine the exact volume fraction and degree of orientation for both helix types independently, a roughly equal mix of α- and $3_1$-helices that are significantly oriented in Z direction is in agreement with both $P_{X/Z} \approx 0$ for amide-I (0.708−0.733 = −0.025) and $P_{X/Z} \approx 0.4$ for amide-III (−0.528 + 0.894 = 0.366); we consider this the most likely scenario (see Supplementary Section 8).

We further carried out [13]C magic-angle spinning nuclear magnetic resonance (MAS NMR) spectroscopy of recluse silk in addition to vibrational spectroscopy, to further analyze its protein secondary structures. Figure 5a shows direct-polarized (purple) and cross-polarized (teal) MAS NMR spectra. Peak assignments were made for the amino acids dominating in spider silks, alanine (Ala) and glycine (Gly) according to previous reports, as detailed in Table 1[1,2,5,58]. Accordingly, resolved peaks for the α-carbons of Ala at 48.9 ppm are well in the range that has previously been associated with β-sheets. The $C_\alpha$-peak of Gly was found at 42.7 ppm, close to what has previously been shown for $3_1$-helices[33,66–68]. The $C_\alpha$-Ala also had a shoulder at 53.3 ppm, close to what would be expected (52.3–52.8 ppm) for α-helical structures. The Ala-$C_\beta$ main peak at 20.4 ppm agrees very well with the

[13]C shifts observed in β-sheet structures, while its shoulder at 16.7 ppm would be typical for random coils. The CO peak at 172.5 ppm, which probably arises from both Ala and Gly residues, falls in the range of both Ala β-sheets and Gly-$3_1$-helices. The CO shoulder at 169.5 ppm is outside the typical range of Ala carbonyl shifts and therefore can be attributed to Gly, as it is a typical shift for a Gly in a β-sheet. Thus, the MAS NMR experiment fully supports our findings from vibrational spectroscopy that the spider silk nanofibrils contain β-sheets, α-helices, $3_1$-helices, and random coil structural features. After determining the presence of the different secondary structures and their orientations, we determined their volumetric percentages. We first calculated the corresponding peak area ratios within amide-I FTIR and Raman bands, as commonly done in the literature, and obtained β-sheet fractions of 41 ± 5% (Supplementary Figure 6 in Supplementary Section 10) and (44.0 ± 4.7)% (Supplementary Figure 7 in Supplementary Section 11), respectively, which is fully in line with our X-ray analysis of this silk, yielding 43% (see Supplementary Figure 5 in Supplementary Section 9). This makes β-crystals the dominating protein secondary structure within *Loxosceles* silk nanofibrils. To the best of our knowledge, 44% is the highest degree of crystallinity observed in spider silk, so far. Previous observations of lower degrees of crystallinity in spider silks might have been caused by the presence of other, non-fibrillar structural elements, like the fiber coating. According to our findings, pristine silk nanofibrils feature a particularly high crystallinity of 44%.

To determine the composition of the remaining secondary structures, we first analyzed the other amide-I Raman sub-peaks

and found 22.1% for type-III β-turns, 8.9% for type-I β-turns, 15.9% for helical structures, and 9.1% random coil. Based on our previous $P_{ZX}$ analysis, α- and $3_1$-helices are present in approximately equal amounts, thus represent ≈8%, each of the nanofibril secondary structure.

For a deeper understanding of the origins of the vibrational peaks in silk nanofibrils, we further considered previously published vibrational spectra of simple polypeptides made from the amino acids dominating in spider silks, Ala and Gly[1,2,5,58]. Poly-alanine (Ala)$_n$ can assume purely α-helical[52,69,70] or purely β-sheet[52,56] confirmations, featuring distinctly different Raman spectra, as shown in Fig. 5b in pink and light blue colors, respectively. Similarly, polyglycine (Gly)$_n$, exists as a purely β-crystalline material ("polyglycine-I"[52,57]) and as a material entirely made of $3_1$-helices ("polyglycine-II"[57,71,72]). For simplicity, we refer to the spectra of these materials as "β-(Gly)$_n$" and "$3_1$-(Gly)$_n$", shown in green and orange colors in Fig. 5b, respectively. For direct comparison with these spectra, we calculated a pseudo-unpolarized Raman spectrum from our polarized spider silk nanofibril spectra (for details see Supplementary Section 11) shown in black in Fig. 5b. All spectra in Fig. 5b feature not only different peak positions, but an entirely different general shape. Nevertheless, we used a least-squares method to fit our nanofibrillar silk spectrum as a mixture of the four polypeptide spectra (all curves normalized for the identical area) and obtained Fit = 0.05·α-(Ala)$_n$ + 0.44·β-(Ala)$_n$ + 0.36·β-(Gly)$_n$ + 0.15·$3_1$-(Gly)$_n$, shown as a dashed, blue line in Fig. 5b; the polypeptide spectra are scaled according to their contributions to the fit. The fit is surprisingly good, given how much simpler the (Ala)$_n$ and (Gly)$_n$ polypeptides are compared to the intricate silk protein with its complicated sequence[58,73–75]. The only major discrepancy is in the spectral region 1550–1650 cm$^{-1}$; however, the peaks in this region are known to be due to side chains of aromatic amino acids rather the protein backbone[47,52,62], and thus could not possibly be present in (Ala)$_n$ and (Gly)$_n$ spectra. The more subtle discrepancy at 1450 cm$^{-1}$ has likely the same origin[52]; other minor deviations are in the 1300–1420 cm$^{-1}$ region (see additional information in Supplementary Section 12). Tables comparing peak positions of our data and these (Ala)$_n$ and (Gly)$_n$ spectra can be found in Supplementary Section 14/15.

Quantitative composition analysis based on the fit coefficients shown above is not expected to be precise, for several reasons. The differences between the sequences of silk vs. the polypeptides are very significant, and the overall spectral intensities of the polypeptides are not calibrated with respect to the amount of material probed; moreover, random coil structures and β-turns are not represented in this approach. Nevertheless, the obtained coefficients show that the fit is strongly dominated by β-sheet contributions from (Ala)$_n$ and (Gly)$_n$, with $3_1$-(Gly)$_n$ and α-(Ala)$_n$ playing minor roles. These findings are well aligned with our vibrational and NMR spectroscopy results. Moreover, the polypeptide spectra provide us with some hints about which amino acids play an important role in the observed secondary structures. According to our fits, β-sheets are dominated by Ala compounds with contributions from Gly; α-helices are dominated by Ala, and $3_1$-helices are mainly Gly. These findings fully support what we concluded from our NMR results.

We propose that polarized vibrational spectroscopy provides great opportunities to further our understanding of silk beyond the significant discoveries discussed above, which is encouraged by the fact that our data and analysis show a high level of consistency. For instance, Raman and FTIR analysis of the amide-I band yielded almost exactly the same fraction of β-sheets. Moreover, almost all peaks associated with β-sheets had $P_{ZX}$ values ≈ ±0.6, whereas all amide-III peaks associated with helical structures featured $P_{ZX}$ ≈ 0.4; random-coil structures, on the other hand, mostly featured $P_{ZX}$ ≈ 0, exactly as expected (see Fig. 4).

There is potential for further exploration of the vibrational peaks in the spectral region between amide-III and amide-II and their orientational preference. The 1366 cm$^{-1}$ and 1450 cm$^{-1}$ IR peaks, for instance, have been suggested to feature both β-sheet and amorphous contributions. Consistently, with these assignments, they feature $P_{ZX}$ values between 0 and 0.6. Understanding these $P_{ZX}$ values as a superposition of 44% β-sheet and 56% amorphous structures, we solved for the $P_{ZX}$ values of these two fractions and obtained values close to 0 and 0.6, respectively, as expected (see Fig. 4a). Furthermore, a more detailed analysis of the assigned sub-peaks in the amide-III band revealed relatively consistent β-sheet vs. helix compositions ratios of 2–3, in agreement with the amide-I Raman analysis (see Supplementary Section 13). The amide-III region also has some currently unassigned peaks, and finally, the strong polarization preference of the peaks due to aromatic side chains (green in Fig. 4) may hold opportunities to learn about their spatial orientation in nanofibrils.

Reaching far beyond the investigation of silk, our procedure and several of our findings will be applicable for many protein/peptide-based materials. As we demonstrated, the combination of p-Raman and p-FTIR spectroscopies with a calculation of the polarization behavior of different secondary structures can provide significant insights into protein structure, even when the material has already been studied exhaustively with standard Raman or FTIR. The discoveries included the determination of volume fractions and orientations of protein secondary structures, and notably, also several previously unknown assignments for peaks in the amide-I/II/III regions, which can be powerful when distinguishing ambiguous secondary structures: (1) the Raman peak at 1658 cm$^{-1}$ had previously been assigned to be due to $3_1$-helices[25], without significant α-helical component[47]. Our polarization analysis showed that this peak contains contributions from both $3_1$-helices and α-helices; (2) we discovered that the Raman peak at 1241 cm$^{-1}$ is due to a mixture of both β-sheet and helical structures, which had not been suggested previously. Such peak assignments are particularly important, because they are universally valid for protein-based materials and can be readily applied—even for spectra taken with standard vibrational spectroscopy. Another concept we have introduced is the idea of superimposing entire spectra of simple polypeptides to explain protein spectra. While prior work had taken into account individual peak positions of polypeptides, we considered linear combinations of entire polypeptide spectra. This removed several ambiguities regarding peak assignments, especially in spectrally populated regions where accidental peak agreement is possible. In particular, this approach also allowed us to specify which amino acids contribute significantly to certain secondary structures, which was previously not straightforward using vibrational spectroscopy. Finally, our generally viable approach will also open the door to better computational approaches, since spectra of relatively simple, periodic polypeptides can be calculated more easily than spectra of complex proteins. In this sense, our method can also bring experimental and modeling approaches to vibrational protein spectroscopy closely together.

We summarize our findings as follows. Using both polarized FTIR and polarized Raman spectroscopy, we revealed the vibrational spectra of purely nanofibrillar, native spider silk. By carefully analyzing the positions, ratios, and polarization preferences of a series of sub-peaks, we concluded that about ≈44% of the nanofibril volume is crystalline β-sheets with a preferential orientation parallel to the fibril axis; the sheets were not perfectly oriented, with alignment angles spreading about ±35°. This is a higher concentration of β-crystals than previous characterizations of entire silk fibers indicated, which may be due to non-fibrillar phases in other silks. We also revealed that spider silk contains two kinds of helical secondary structures, α-helices, and

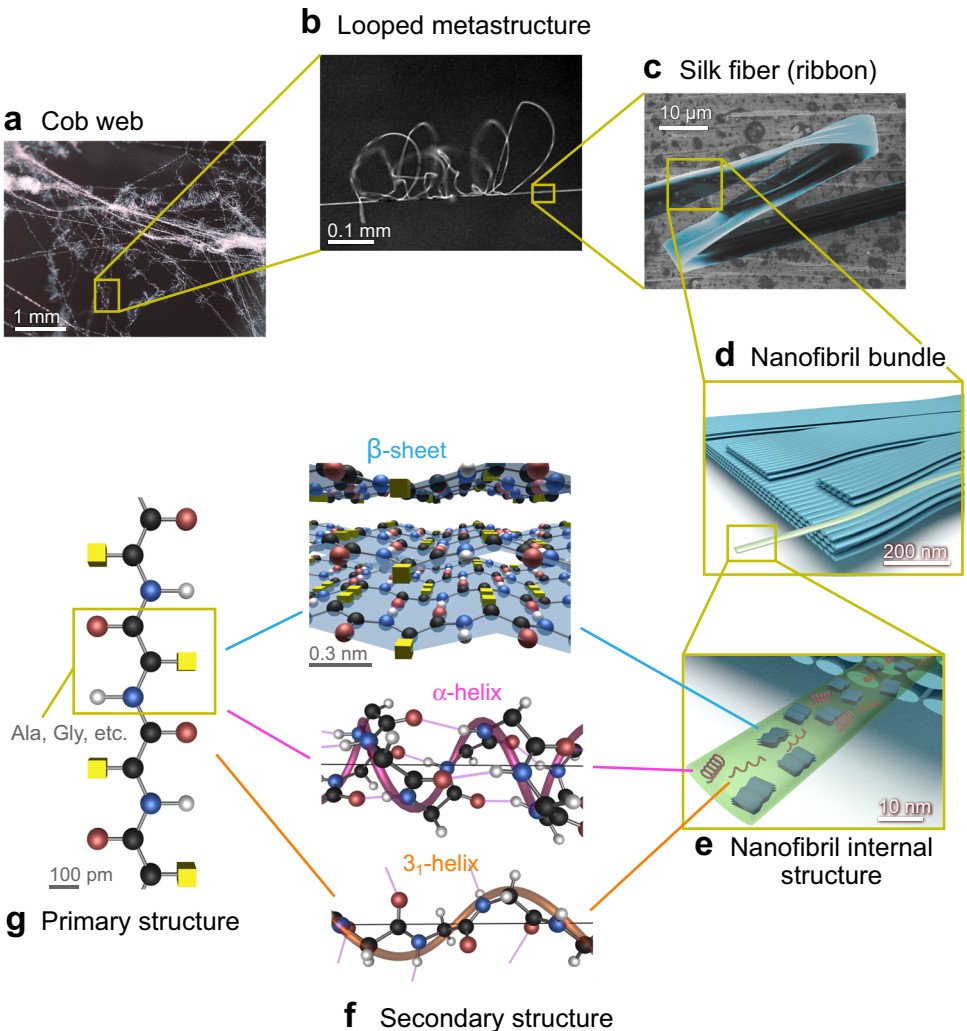

**Fig. 6 Hierarchical structure of Loxosceles silk. a** The recluse spider's cob-web. **b** Looped metastructure made of *Loxosceles* silk ribbons. **c** Single ribbon (false-colored SEM, $n = 5$ independent SEM experiments). **d** Schematic of ribbon entirely composed of 20 nm thin nanofibrils. **e** β-sheets (blue), α- (pink) and 3₁-helices (orange) within a single nanofibril (schematic). **f** Ala-β-sheet, Gly-3₁-helix, and α-helix segment in protein 2N3E[83]. Yellow cubes: functional groups on the $C_\alpha$ atoms; transparent spirals: approximate helix structure. The color scheme used for different atoms are the same as Fig. 3. **g** Silk protein primary structure.

$3_1$-helices, both featuring an orientation preference parallel to the fiber axis. These findings were in full agreement with our MAS NMR data. Finally, we were able to approximate our measured silk spectra by a superposition of polypeptide spectra taken exclusively from the amino acids dominating spider silk, Ala and Gly. This hints that the vibrational spectra of the complicated silk proteins can be understood in terms of much simpler peptide structures, providing promising avenues to explore quantitative modeling efforts based on first principles. The polypeptide fitting approach, and our NMR MAS data indicated that Ala dominates α-helices and plays the main role in β-sheets; Gly dominates $3_1$-helices and plays an important role in β-sheets. Our approaches have great potential to improve the understanding of the secondary structures and orientations of other proteins and protein-based materials, and to further our understanding of their vibrational spectra.

Based on this summary and on our previous works[6,24], we propose a detailed structural model of *Loxosceles* silk in Fig. 6, featuring 7 levels of hierarchy. The recluse spider makes a cob-web construction for its silk ribbons (Fig. 6a), where a looped metastructure can be found on a single silk strand (Fig. 6b). This 6–8 μm wide, ≈50 nm thick ribbon silk (Fig. 6c) is entirely composed of 20 nm thin nanofibrils (Fig. 6d)[6]. These silk nanofibrils are mostly composed of β-sheets, with additional contributions from $3_1$- and α-helices and random coil (Fig. 6e–f). This model is more complete than other spider silk models in the literature, in which the fraction of nanofibrils and potentially other structural components (Fig. 6d) has not yet been determined, and for which the secondary structure (Fig. 6e, f) has not yet been revealed with this degree of accuracy. Our work provides composition estimates for six different protein secondary structures in the nanofibrils: β-sheets, type-I β-turns, type-III β-turns, $3_1$-helices, α-helices, and random coil. In combination with available mechanical property data for this silk[6,8,24], this wealth of structural data will provide great opportunities to develop more rigorous structure–property relationships of spiders silk, which will ultimately guide the development of high-performance synthetic fibers inspired by spider silk.

## Methods

**Sample preparation**. For silk harvesting, adult *Loxosceles laeta* spiders were first anesthetized with $CO_2$ and then restrained. For FTIR measurements, silk strands were reeled in 3–5 turns over a 1 mm diameter aperture in a rectangular aluminum substrate at a steady rate of ≈1 cm/s (Fig. 1d). This gave rise to fiber separations

large enough to enable single-strand IR probing. For the Raman samples, strands of *Loxosceles* MA silk collected from the cotton strip inside the capsule were carefully deposited on a TEM grid (Ted Pella, 90 μm hole width) to prevent any excessive tension applied to the silk ribbons. The TEM grid also improved the stability of freely suspended silk strands. Locations where the silk ribbon was suspended flat over the hole were chosen to be examined (Fig. 1e).

**Fourier-transform infrared (FTIR) spectroscopy**. To carry out the single-strand FTIR measurement, a Bruker 80 v FTIR vacuum spectrometer was coupled to a Spectratech optical microscope with a confocal Schwarzschild objective of numerical aperture 0.58. The whole microscope was encased in a chamber filled with ≈10 psi of compressed air. This compressed air had been scrubbed of $CO_2$ and $H_2O$ by our Parker Balston gas purge in order to eliminate the $CO_2$ and $H_2O$ absorption lines in the mid-infrared spectral range. Data was taken with confocal rectangular 0.25 mm × 3 mm apertures. With a ×15 magnification, this equates to a ≈16 μm × 200 μm spot at the focus of the object. This aperture size was chosen to maximize the area of the single-strand in the field of view. It was natural to elongate the apertures along the length of the strand while reducing the width, which further allowed us to get sufficient signal-to-noise while maintaining single-strand precision.

We used a water-cooled, high temperature blackbody (globar) as the infrared (IR) light source (Supplementary Fig. 1). A polarizer consisting of a wire grid on KRS-5 substrate (Optometrics) was placed in the path of the incident beam. The infrared light was polarized parallel and perpendicular to the long axis of the silk strand by manually rotating the polarizer. Data was taken between 1000 cm$^{-1}$ and 4000 cm$^{-1}$ with a 4 cm$^{-1}$ spectral resolution using a KBr beamsplitter and a liquid nitrogen cooled mercury-cadmium-telluride (MCT) photodetector. Moveable sample and detector stages were designed and implemented for optimized precision transmission measurements.

The infrared transmission data through the spider silk sample was normalized relative to the transmission through an aperture of the same size as that used for the silk. A frequency dependent background exists in the transmission spectrum, which likely occurs as a result of frequency dependent diffraction, reflectance and scattering due to the geometry of the spider silk fiber used for transmission measurements. This background was removed using the following procedure. We initially excluded the spectral range of infrared vibrational modes and used a polynomial to fit the remaining featureless spectrum which was mainly attributed to transmission background. We then subtracted this polynomial fit from the raw transmission data to obtain a background-free transmission spectrum. Then, the transmission spectrum was converted to an absorbance spectrum via the Beer-Lambert Law: $A = -\log \frac{I}{I_0}$, where $I_0$ is the intensity of the beam transmitted through the reference aperture (i.e., free space), and $I$ is the intensity of the beam transmitted through only the silk strand.

**Raman spectroscopy**. Raman spectra were collected using a Renishaw inVia Raman Microscope/Spectrometer system. An internally polarized green laser line (Laser Physics Inc., West Jordan, UT) with a wavelength of 514 nm was used to excite the sample. All measurements were done with a Leica ×100 objective (0.90 N.A.). For the 500–4000 cm$^{-1}$ measurement, an exposure time of 60 s was used. For the 1150–1750 cm$^{-1}$ measurements, we increased the exposure time to 900 seconds to reduce noise. To optimize the mechanical stability, we used a 3-axis micro-manipulator (Siskiyou Corporation, Grants Pass, OR) to hold our sample. In order to obtain the p-Raman spectra, a film linear polarizer (Thorlabs) was used as the analyzer and positioned either parallel or perpendicular to the polarization direction of the incident laser beam. The orientation of the sample was changed accordingly for different polarization configurations. A linear background was subtracted for some of the spectra when needed.

A simplified scheme showing the Raman and FTIR setup is featured in Supplementary Fig. 1. The linear polarization directions are either parallel ("Z") or perpendicular ("X") to the ribbon axial direction, which is also the nanofibril direction. For the Raman measurements, an analyzer was used before the detector to select scattered light with different polarizations.

**Amide band deconvolution**. For curve-fitting of Raman spectra, peaks with mixed Gaussian/Lorentzian characteristics were used[25,76], with the following four fit parameters for each peak: peak position, peak height, full width at half maximum (FWHM), and the percentage of Lorentzian character. Following common literature, five peaks[25] were used to fit the amide-I band, three peaks[47] for the amide-III band, and other peaks at positions previously suggested[47,77] for the remaining range. For the initial peak positions and FWHMs of the fitting process for each peak we used data from the literature[25,47,77]. The initial degree of Lorentzian character was randomly chosen for each peak, and the initial peak magnitudes were estimated based on the overall spectral intensity. Furthermore, the peak position, FWHM, and percentage of Lorentzian character were confined to be identical for the same peak across different Raman spectra. Finally, the least-square method was used for the fitting; the fitting process was carried out using MATLAB.

In order to test the robustness and stability of our decomposition method, we introduced uniformly distributed errors to the initial fitting parameters of each peak.

Multiple fitting processes ($n_F = 15$) were performed for the Raman spectra. The maximum variations of the randomly generated errors for the peak position, FWHM, peak magnitude, and Lorentzian content were limited to ±5 cm$^{-1}$, ±5 cm$^{-1}$, 10%, and 0%–100%, respectively. The fitting results showed a high level of stability, suggesting that our fits are very good approximations of the experimental spectra.

For the FTIR spectra, the same general strategy was followed, using Kramers–Kronig consistent Lorentz oscillators in the dielectric function for fitting the experimental transmission data. The fits were done using W-VASE software from J.A. Woollam, Inc. The transmission due to individual oscillators was then converted to absorbance. To estimate the uncertainty in the fitted peaks, we performed constrained fits of the amplitude and bandwidth of the oscillators. Based on our error bounds and agreement with previously referenced peak positions, we have high confidence in the uniqueness of our fits (see Supplementary Section 5).

**NMR**. For NMR experiments, ~40 mg of silk collected from both male and female *Loxosceles laeta* spiders were packed into a 4 mm zirconia rotor (Bruker). MAS NMR experiments were performed on a 17.6 T (750 MHz) Avance I wide bore (Bruker, Billerica, MA) spectrometer using a Bruker BL4 HXY 4 mm MAS probe (tunable to $^1$H-$^{13}$C-$^{15}$N). Spinning was regulated at 12.5 kHz using a Bruker MAS II pneumatic MAS controller. All experiments were performed with a variable temperature set to 25 °C with a Bruker BVT-3000 temperature controller. $^{13}$C experiments employed either direct-polarization (DP) or ramped cross-polarization (CP)[78]. All experiments employed SPINAL-64 1H decoupling[79] during acquisition, with a nutation frequency of ≈63 kHz. Recycle delays were 1.5 and 10 s for CP and DP $^{13}$C spectra, respectively. The contact time for the CP experiment was 2 ms. Spectra were externally referenced to adamantane, assuming the downfield peak at 38.48 ppm[80]. NMR spectra were processed with 50 Hz apodization and zero filled prior to Fourier transformation.

**Collection of published model polypeptides spectra**. To digitize the polarized or unpolarized IR and Raman spectra of several model polypeptides published before, we first cleaned up the collected figures with G.I.M.P. (https://www.gimp.org/). This step usually involved changing the figure to gray scale, cropping out unwanted portions, and removing the noisy background and unnecessary legends. Then we used Engauge Digitizer (http://markummitchell.github.io/engauge-digitizer/) to translate the graphical curve to digital data points.

**Reporting summary**. Further information on research design is available in the Nature Research Reporting Summary linked to this article.

## Data availability

The IR, Raman, NMR, and XRD data generated in this study and the codes used for data analysis have been deposited in the Harvard Dataverse under accession code (dataset persistent ID) https://doi.org/10.7910/DVN/GMEEYL[81]. Data are also available from the corresponding author upon request.

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

## Acknowledgements

H.C.S. acknowledges funding from the National Science Foundation (NSF) under Grant Nos. DMR-1905902, DMR-1352542, and DMR-2105158. M.M.Q. acknowledges support from NSF under Grant No. IIP-1827536. P.M., S.L.W., and M.M.Q. are grateful for support from the Virginia Space Grant Consortium (VSGC) fellowship program.

## Author contributions

Q.W. prepared all silk samples and carried out Raman experiments and pertinent data analysis; P.M., S.L.W., R.L.W., and Z.X. carried out IR experiments and analyzed the data; M.M.Q. supervised I.R. experiments and interpretation; A.G. and M.L.C. planned and interpreted the MAS NMR experiments, with A.G. collecting the data; Q.W. and H.C.S. wrote the manuscript and developed the overarching model and interpretation.

## Competing interests

The authors declare no competing interests.
