## [Peer Review File · Nature Communications]

Protein Secondary Structure in Spider Silk NanofibrilsREVIEWER COMMENTS

Reviewer #1 (Remarks to the Author):

The article untitled « Protein Secondary Structure in Spider Silk Nanofibrils » (ref. NCOMMS-21-39575) is devoted to the study of the molecular structure of spider silk using a combination of FTIR, Raman and NMR spectroscopies. The objective is to obtain a description as detailed as possible of the molecular structure of a nanofibril, the basic unit that forms silk fibrils of the recluse spider.

The strength of the work is the use a combination of complementary techniques (FTIR and Raman) with an in-depth analysis of various vibrational modes (amide I, amide II and amide III). Although some aspects remain to be confirmed, the image of the structure that is obtained reaches a very interesting level of detail. The procedure developed to obtain the infrared polarized spectra of silk appears promising. Several spectroscopic arguments and calculations (i.e., reconstruction of experimental spectra with model peptides) provide very interesting insights.

The manuscript is completed by exhaustive, appropriate and up-to-date references. It is well-written and organized and the figures are clear. The article then deserves publication, but I am not convinced that, for the following reasons, Nature Communications is the appropriate journal.

First, I am not sure that the conclusions and the analysis would interest a sufficiently large audience. The methodology and results are very interesting to the silk community but in my opinion there is no breakthrough conclusion, especially for (bio)materials scientists.

Second, the novelty of the work (investigating nanofibrils) appears to me lower than claimed by the authors. Indeed, the techniques used do not have the spatial resolution to probe individual nanofibrils. Thus, although the fibre is composed of nanofibrils, the data cannot provide information regarding individual nanofibers. In my opinion, the phrase “spider silk nanofibrils” used throughout the text does not correspond to the work. As a consequence I suggest changing the title and the claimed novelty of this work.

Third, the “unpolarised” Raman and FTIR spectra have not been appropriately calculated, which questions the percentages of secondary structures. For IR, it is not sufficient to use unpolarised light since the sample is isotropic. The unpolarised spectrum is well known to be given by: $(|Z| + 2|X|)/3$. For Raman, the Eq. in S11 is not appropriate. To calculate “unpolarised” spectra, one should use the expression of rotational invariant spectra (Frisk et al. Applied Spectroscopy (2003) 57(9)-1053), especially taking into account depolarization effects (Lefevre et al. Applied Spectroscopy (2006) 60(8)-841).

The spectral curve-fitting procedure should be detailed: there exist multiple mathematical solutions to decompose the spectra, especially on a so broad wavenumbers region. How the authors can be confident that their calculation is (close to) the real solution? Many details are lacking regarding the curve-fitting procedure: in particular, what was the degree of freedom of the spectral parameters (bandwidth, position). Were they totally free to evolve or were their values restricted? Also, how was chosen the initial parameters (before parameter adjustment)?

Reviewer #2 (Remarks to the Author):

This article uses polarized FTIR and polarized Raman spectroscopy to analysis the secondary structure of ribbon silk of the recluse spider. As described in the manuscript, by using the recluse spider silk, the authors can overcome the difficulties in analyzing the secondary structure of nanofibrils when they cannot be isolated, which featuring a mixture of different secondary structures and may have a great interfere to the obtained spectra. The authors provide a complete set of methods. Through the polarization dependency of the protein structure, people can have a deeper understanding of the details of the secondary structure from the aspects of symmetry and develop a structural model with unprecedented detail, featuring seven levels of structural hierarchy for the recluse spider. Overall, this work is interesting and is innovative. The results are well presented. I only have several minor suggestions.

1. Both polarized Raman and polarized Fourier transform infrared (p-FTIR) spectroscopies have been used to investigate the secondary structures and orientations in spider silks and silkworm silks, which

are also very commonly combined with MAS NMR and synchrotron FTIR (S-FTIR). More comparisons with these works are suggested to reveal the significance of this work in introduction.

2. Some descriptions seem exaggerated. "We carried out polarized Raman and polarized Fourier-transform infrared spectroscopies on native spider silk nanofibrils for the first time..." (native spider silk nanofibrils have been commonly analyzed by these two methods); "Although silk nanofibrils have been observed in various silks, their prevalence in silk fibers was completely unclear until recently, when we demonstrated that the strong and tough ribbon silk of the recluse spider..." (The existence of microfibrils in silk should have been very clear long ago)

3. Does the method introduced in this paper only work for this specific spider silk? Does the structure of this ribbon silk of the recluse spider differ greatly from most other spider silk? How much reference can the model introduced in this work provide for other kinds of spider silk?

4. What's the importance to provide the optical microscope pictures of the infrared and Raman experiments in Figure 1? maybe replaced with a picture of the spider silk ribbon under a polarizing microscope.

5. To provide a better understanding for a wider readership, a brief description is suggested to include: What is the difference between Raman spectroscopy and FTIR? Do these two characterization methods complement or verify the structural information?

Reviewer #3 (Remarks to the Author):

The manuscript deals with the critical issue of secondary structure determination in silk fibres. As rightly pointed by the authors, this is a topic highly researched and debated. The merit of the manuscript is in lifting the ambiguity often encountered during secondary structure assignments. The work presented is exhaustive and well-grounded. The use of complementary methods and polarisation to enhance the information content.

A few general comments and questions

1. Silk fibres are often co-spun. Do the authors anticipate contribution from, for example, aciniform or piriform silk onto the significant ampullate silk under investigation?

2. Vibrational spectroscopies (FTIR and Raman) are local probes. The direct consequence is that the size of the secondary structure may be unattainable. Therefore, the question is: Can the secondary structure's length cause some misinterpretation of the peak assignments?

3. Sample under investigation. It is unclear if the size of silk fibre/fibril under measurement is the same for the FTIR and Raman?

Specific questions

Line 35: "that the outstanding mechanical properties of spider silk are already implemented at the level of an individual nanofibril". The statement is unclear. If referring to a tensile test, it will depend on the region considered. In a way, the purpose of the paper is to substantiate the statement.

Line 237: "However, since α - and 3_1 -helices show opposite polarization preference, an alternate scenario leading to $PZX \approx 0$ is a mixture of oriented α - and 3_1 -helices. Considering the amide-I peak alone, the two scenarios cannot be distinguished." The results are remarkable and highlight the strength of the polarisation analysis. However, this assumes that the absorption cross-section of the two types of helices is similar, is that true? or that the two helices are of similar sizes and, therefore, contribution?

RESPONSES TO THE REVIEWER COMMENTS

The authors would like to thank all the reviewers of this manuscript for their thoughtful feedback and for the time they spent, which helped to improve our manuscript significantly. Our detailed responses to all issues raised by the reviewers are provided in blue color below.

Reviewer #1 (Remarks to the Author):

The article untitled « Protein Secondary Structure in Spider Silk Nanofibrils » (ref. NCOMMS-21-39575) is devoted to the study of the molecular structure of spider silk using a combination of FTIR, Raman and NMR spectroscopies. The objective is to obtain a description as detailed as possible of the molecular structure of a nanofibril, the basic unit that forms silk fibrils of the recluse spider.

The strength of the work is the use a combination of complementary techniques (FTIR and Raman) with an in-depth analysis of various vibrational modes (amide I, amide II and amide III). Although some aspects remain to be confirmed, the image of the structure that is obtained reaches a very interesting level of detail. The procedure developed to obtain the infrared polarized spectra of silk appears promising. Several spectroscopic arguments and calculations (i.e., reconstruction of experimental spectra with model peptides) provide very interesting insights.

#1.1 Response: We thank the reviewer for the positive feedback.

The manuscript is completed by exhaustive, appropriate and up-to-date references. It is well-written and organized and the figures are clear. The article then deserves publication, but I am not convinced that, for the following reasons, Nature Communications is the appropriate journal.

First, I am not sure that the conclusions and the analysis would interest a sufficiently large audience. The methodology and results are very interesting to the silk community but in my opinion there is no breakthrough conclusion, especially for (bio)materials scientists.

#1.2 Response: We thank the reviewer for this feedback. We agree with the reviewer that the significant impact of our work beyond the silk field had not been sufficiently explained in our original manuscript. We took the opportunity to substantially strengthen our manuscript in this regard by adding a substantial paragraph at the end of our Results and Discussion section, as well as small additions to the conclusions and abstract. This will make the importance of our work for the wide audience of *Nature Communications* much clearer. As explained in the revised manuscript, these are the breakthrough conclusions of our work, which are especially interesting for (bio)materials scientists. Most importantly:

1. The procedure we described is generally viable for any protein/peptide-based material and not just for silk. It dramatically increases the amount of information accessible from a sample via vibrational spectroscopy. While many current studies simply perform Raman or Fourier-transform infrared (FTIR) spectroscopy, we combined polarized Raman spectroscopy and polarized FTIR of a well-defined sample with a calculation of the polarization behavior of different secondary structures, followed by detailed analysis. From this information we gained a much deeper understanding of the sample. Notably, we determined volume fractions of secondary

structures that were previously not available, even though they had been studied exhaustively with techniques that were considered state of the art. Moreover, we also determined the spatial orientations of most of these structures. Notably, we were also able to make several new assignments for peaks in amide I/II/III regions. The latter is particularly important, because once these assignments are made for a particular material, future studies become more powerful, even when carried out with simpler equipment, such as widely available standard Raman spectroscopy.

2. A second new concept we introduced here is the idea of superimposing entire spectra of simple polypeptides to explain the spectrum of a protein. Alanine and glycine play an important role in spider silk, and thus we have considered the spectra of polyalanine and polyglycine. While prior work had taken into account individual peak positions of polypeptides, we considered linear combinations of entire polypeptide *spectra*. This removed several ambiguities regarding peak assignments, especially in spectrally populated regions where accidental peak agreement is possible. In particular, this approach also allowed us to specify which amino acids had a significant contribution certain secondary structures, which was previously not possible using vibrational spectroscopy. This will also open the door to better computational approaches, since spectra of relatively simple, periodic polypeptide can be calculated more easily than spectra of complex proteins. Therefore, our approach also provides an opportunity to bring experiment and modeling closer together.

Our approach is viable for any proteinaceous material, for instance the important class of collagen-based materials, including cartilage, skin, and bone. Consequently, it is clear that our work has significant impact far beyond the silk field, on the basis of our breakthroughs regarding improved peak assignment, deeper insights into structural organization and orientation, revelation which amino acids contribute to a particular protein secondary structure, and the opportunity to improve computational models for protein spectra. Based on this broad impact, we believe that our findings will be interesting to a significant fraction of the broad readership of *Nature Communications*., and our revised manuscript articulates this clearly.

Second, the novelty of the work (investigating nanofibrils) appears to me lower than claimed by the authors. Indeed, the techniques used do not have the spatial resolution to probe individual nanofibrils. Thus, although the fibre is composed of nanofibrils, the data cannot provide information regarding individual nanofibers. In my opinion, the phrase “spider silk nanofibrils” used throughout the text does not correspond to the work. As a consequence I suggest changing the title and the claimed novelty of this work.

#1.3 Response: The reviewer is correct that we did not take spectra from individual nanofibrils — and we certainly do not claim this anywhere in the manuscript. However, as we will explain in the following, we do not think that the question whether spectra are taken from individual nanofibrils, or from a pure bundle consisting exclusively of nanofibrils makes a significant difference for the claims, conclusions, or novelty of our work. Notwithstanding if single or multiple nanofibrils were sampled, our manuscript is still the first work reporting vibrational spectra of native spider silk nanofibrils. This is important, because nanofibrils are generally considered a critically important constituent of spider silk.^{R1} However, for all other spider silks we are aware of, non-fibrillar components have been detected or proposed, such as a matrix of amorphous silk surrounding the nanofibrils,^{R1,R2} silk protein in globular or micellar form,^{R3} as well as glycoprotein and lipid coatings of the fiber.^{R4,R5} We appreciate the reviewer’s feedback, because it

has caused us to realize that our motivation and discussion of these circumstances was too abbreviated in the original manuscript. We have corrected this and added this information along with references in the introduction in the revised version of our manuscript. The described circumstances have, so far, prevented the acquisition of spectra from nanofibrils only. We have succeeded in doing just that, which **has allowed us to determine the secondary structure of native spider silk nanofibrils for the first time. For spectra taken from all other spider silk fibers this cannot be established, because it would never be clear whether the signal is actually from nanofibrils, or from other, non-fibrillar constituents.**

At this stage, we do not have any reason to believe that the spectrum of an individual nanofibril were different from the spectrum of a nanofibril bundle. Consequently, our claims and conclusions stand. Spider silk consists of a small number of proteins with a well-defined sequence, and therefore, inhomogeneities between different nanofibrils are not expected — and we have no indication of any such inhomogeneities whatsoever. The Raman/FTIR spectra of nanofibrils are defined by molecular vibrations of the protein molecules within a nanofibril. We doubt that the presence of other nanofibrils were to alter these vibrations in a significant manner. From a practical point of view — aiming for identification of structures in silk fibers — spectra of purely nanofibrillar bundles may be as important (or more important) than spectra of individual nanofibrils.

The important distinction and novelty of our work is that we analyzed the spectra and determined the structure of nanofibrils, as opposed to other, not nanofibrillar constituents of spider silk. Therefore, we think it is important to reflect this important distinction in the title of the manuscript. However, the reviewer's question has shown us that our presentation of this issue in the original manuscript left room for ambiguity and did not explain the subtleties of this matter in enough detail. Correspondingly, we have made several changes throughout the manuscript that we believe now fully address the reviewer's concerns, in particular we made sure none of our statements could be understood as a claim that we characterized individual nanofibrils.

Third, the “unpolarised” Raman and FTIR spectra have not been appropriately calculated, which questions the percentages of secondary structures. For IR, it is not sufficient to use unpolarised light since the sample is isotropic. The unpolarised spectrum is well known to be given by: $(|Z| + 2|X|)/3$. For Raman, the Eq. in S11 is not appropriate. To calculate “unpolarised” spectra, one should use the expression of rotational invariant spectra (Frisk et al. Applied Spectroscopy (2003) 57(9)-1053), especially taking into account depolarization effects (Lefevre et al. Applied Spectroscopy (2006) 60(8)-841).

#1.4 Response: The methods to calculate unpolarized Raman and FTIR spectra proposed by the reviewer (Frisk et al.^{R6}) were indeed used successfully for fibers with **cylindrical** morphology. These methods explicitly assume uniaxial symmetry, where $|X|=|Y|$, which indeed leads to $(|Z|+2|X|)/3$ for unpolarized FTIR spectra. However, the recluse spider silk fiber is a ribbon with rectangular cross section, and therefore, the symmetry reduction due to cylindrical geometry does not apply. **To our knowledge, this is the first time a silk ribbon with rectangular cross has been measured with IR spectroscopy.** The thickness of the silk ribbon (Y direction) is about 50 nm which is much smaller than both the width (X direction) of about 10 μm and the length (Z direction) of the order of millimeters. Therefore, the ribbon can be approximated as a **quasi-two-dimensional material with pseudo-layered structure** (Wang and Schniepp, Ref [R7]) for the purpose of Raman and IR measurements and analysis to obtain percentages of

secondary structures. In our experiments, the incident wave-vector of light is propagating at near-normal incidence to the XZ surface of the quasi-two-dimensional silk ribbon. For the IR, we obtained the data with the electric field polarized in the X - and Z -directions and then calculated the (pseudo)unpolarized spectrum in the XZ plane of the silk as $(|Z|+|X|)/2$, as suitable given the quasi-two-dimensional geometry of the silk ribbons. For the Raman data, the calculation for this orthotropic symmetry is more complicated, as there can be up to 16 unknown variables^{R8} — many more than the 5 variables needed for the uniaxial symmetry employed by Frisk et al.^{R6} A mathematically rigorous derivation of the fully correct unpolarized spectrum is challenging and would be beyond the scope of this manuscript. Our approximation uses the sum of all nine Raman spectra with different polarization configurations to represent the (pseudo) unpolarized Raman spectrum. In essence, this aligns with the experimental observable part of equation (13) derived by Frisk et al.^{R6} We believe that this approximation to determined unpolarized Raman spectra is sufficient in our case, especially since the obtained agreement between Raman (44%), FTIR (41%) and X-ray diffraction (43.2%) is satisfactory.

We have expanded the Supplementary Information document regarding the calculation of pseudo-unpolarized Raman and FTIR spectra. We now explain in greater detail the implication of the symmetry, and why the methods used for cylindrical fibers do not apply for the silk ribbons studied here.

The spectral curve-fitting procedure should be detailed: there exist multiple mathematical solutions to decompose the spectra, especially on a so broad wavenumbers region. How the authors can be confident that their calculation is (close to) the real solution? Many details are lacking regarding the curve-fitting procedure: in particular, what was the degree of freedom of the spectral parameters (bandwidth, position). Were they totally free to evolve or were their values restricted? Also, how was chosen the initial parameters (before parameter adjustment)?

#1.5 Response: Following the suggestion of the reviewer, we happily added more details regarding our curve-fitting procedure to the “Amide band deconvolution” part of the manuscript’s Methods section, which now reads as follows:

“For curve-fitting of Raman spectra, peaks with mixed Gaussian/Lorentzian characteristics were used, with the following four fit parameters for each peak: peak position, peak height, full width at half maximum (FWHM), and the percentage of Lorentzian character. Following common literature, five peaks were used to fit the amide-I band, three peaks for the amide-III band, and other peaks at positions previously suggested for the remaining range. For the initial peak positions and FWHMs of the fitting process for each peak, we used data from the literature. The initial percentage of Lorentzian character was randomly chosen for each peak, and the initial peak magnitudes were estimated based on the overall spectral intensity. Furthermore, the peak position, FWHM, and percentage of Lorentzian character were confined to be identical for the same peak across different Raman spectra. Finally, the least-square method was used for the fitting; the fitting process was carried out using MATLAB.

In order to test the robustness and stability of our decomposition method, we introduced uniformly distributed errors to the initial fitting parameters of each peak. Multiple fitting processes ($n = 15$) were performed for the Raman spectra. The maximum variations of the randomly generated errors for the peak position, FWHM, peak magnitude, and Lorentzian content were limited to $\pm 5 \text{ cm}^{-1}$, $\pm 5 \text{ cm}^{-1}$, 10%, and 0%–100%, respectively. The fitting results showed a high level of stability, which caused us to believe our fits are very good approximations of the experimental spectra.

For the FTIR spectra, the same general strategy was followed, using Kramers–Kronig consistent Lorentz oscillators in the dielectric function for fitting the experimental transmission data. The fits were done using W-VASE software from J. A. Woollam, Inc. The transmission due to individual oscillators was then converted to absorbance. To estimate the uncertainty in the fitted peaks, we performed constrained fits of the amplitude and bandwidth of the oscillators. Based on our error bounds and agreement with previously referenced peak positions, we have high confidence in the uniqueness of our fits (see Section S5 in Supplementary Information).”

We believe that this additional information in the manuscript addresses all questions raised by the reviewer. We also note that all codes used for the fits are available in the data package accessible for the reviewers (and for the readers at the point of publication).

Reviewer #2 (Remarks to the Author):

This article uses polarized FTIR and polarized Raman spectroscopy to analysis the secondary structure of ribbon silk of the recluse spider. As described in the manuscript, by using the recluse spider silk, the authors can overcome the difficulties in analyzing the secondary structure of nanofibrils when they cannot be isolated, which featuring a mixture of different secondary structures and may have a great interfere to the obtained spectra. The authors provide a complete set of methods. Through the polarization dependency of the protein structure, people can have a deeper understanding of the details of the secondary structure from the aspects of symmetry and develop a structural model with unprecedented detail, featuring seven levels of structural hierarchy for the recluse spider. Overall, this work is interesting and is innovative. The results are well presented. I only have several minor suggestions.

#2.0 Response: We thank the reviewer for their positive feedback.

1. Both polarized Raman and polarized Fourier transform infrared (p-FTIR) spectroscopies have been used to investigate the secondary structures and orientations in spider silks and silkworm silks, which are also very commonly combined with MAS NMR and synchrotron FTIR (S-FTIR). More comparisons with these works are suggested to reveal the significant of this work in introduction.

#2.1 Response: We appreciate the reviewer’s suggestions and have made additions in our revised manuscript for each of the suggested topics. Regarding the combination with MAS NMR, we have added the following paragraph to the manuscript (reference numbers correspond to the bibliography of the manuscript):

“Previous MAS NMR studies of ^{13}C -labelled spider silk³³ from *Nephila clavipes* have suggested that the Gly-Gly-Ala motif forms disordered 3_1 -helices while the poly-Ala and poly-(Gly-Ala) regions adopt an ordered β -sheet structure. Similar studies³⁴ on dragline silk from *Argiope aurantia* specifically $^{13}\text{C}/^{15}\text{N}$ -labelled on proline residues indicated that the Gly-Pro-Gly-X-X motif, found only in major ampullate spidroin 2 (MaSp2), adopts an elastin-like β -turn structure. Subsequently, further studies³⁵ found additional evidence for these structures as well as evidence for some α -helical structures in the poly-Gln-Gln-Ala-Tyr regions in the dragline silk of *Latrodectus hesperus* (black widow). These conclusions were all drawn from chemical shift information obtained from MAS NMR, either cross-polarization (CP) 1D⁴⁰ or 2D ^{13}C - ^{13}C DARR⁴¹ or INADEQUATE⁴² spectra or in the case of $^{13}\text{C}/^{15}\text{N}$ -labelled proline, $^{15}\text{N}/^{13}\text{C}$ HETCOR spectra.⁴³”

Regarding the use of p-FTIR and S-FTIR, we have added the following:

“Polarized Raman (p-Raman)^{36,47} and polarized FTIR (p-FTIR)^{38,48–50} have been separately applied to investigate the secondary structures of natural cylindrical spider silks and silkworm silks. Both the qualification and quantification of various structural elements have been examined. Similarly, with its high signal-to-noise ratio, synchrotron FTIR (S-FTIR) was also implemented to study the protein secondary structure in natural silk fibers.^{37,51} However, due to the complex structural makeup of cylindrical spider silks and quasi-cylindrical silkworm silks, no direct spectroscopic information has been available for pristine silk nanofibrils.”

2. Some descriptions seem exaggerated. “We carried out polarized Raman and polarized Fourier-transform infrared spectroscopies on native spider silk nanofibrils for the first time...” (native spider silk nanofibrils have been commonly analyzed by these two methods); “Although silk nanofibrils have been observed in various silks, their prevalence in silk fibers was completely unclear until recently, when we demonstrated that the strong and tough ribbon silk of the recluse spider...” (The existence of microfibrils in silk should have been very clear long ago)

#2.2 Response: As already expressed in our #1.3 Response above, we took the reviewer feedback as an opportunity to improve our manuscript. Our explanations in the original manuscript may have been too brief for readers unfamiliar with the state of the art of spider silk nanofibril investigations. We believe that the revised manuscript provides more context and detail to show that our statements are fully supported on the basis of all published evidence. We are also happy to provide as many details as needed here. As a matter of fact, we previously devoted an entire review article to the very question of what is known and what is not known about nanofibrils in silk fibers.^{R9} For the latter article, which was well received by the community, we reviewed the entirety of the reported experimental evidence concerning nanofibrils in silk fibers from spiders and silkworms. After our thorough review we concluded that simply no one in the literature has claimed to have prepared native spider silk nanofibrils in pure form. Therefore, all prior publications showed spectra of materials that likely contained nanofibrils, albeit at an unknown concentration! However, that means that **previously, it was not clear, which contributions the fibrillar and the non-fibrillar components made to the vibrational spectra of silk**. In other words, it has not been established what the spectra of native spider silk nanofibrils look like, and therefore our claim in this manuscript.

The reviewer is correct that there have long been hints about nanofibrillar elements in silk fibers in the published evidence. Importantly, however, the great majority of the proposed silk fiber models do include non-fibrillar components, such as surrounding amorphous matrices,^{R1,R2} as well as glycoprotein and lipid coatings.^{R4,R5} To the best of our knowledge, there is not a simple report that would establish the volume fraction of nanofibrils in a silk fiber.^{R9} The vibrational spectra of native silk in the literature were taken from silk *fibers*; however, because we do not know whether 10% or 90% of these fibers actually consist of *nanofibrils*, the spectra of nanofibrils cannot possibly be obtained or derived from this data.

The reviewer further states “The existence of microfibrils in silk should have been very clear long ago”. We actually agree with the reviewer on this statement. However, we are not claiming to be the first team to find fibrils in spider silk. What we are claiming that for the first time we have a silk fiber that consists of nanofibrils, and of nanofibrils *only*. This is critically important, because as discussed above, this is the only way we can determine the vibrational spectrum of a silk nanofibril.

We made changes to the text reflecting these arguments and this point of view throughout our revised manuscript to explain these issues more clearly and to provide a more nuanced representation. In particular, we added information in the introduction highlighting the community consensus that non-fibrillar components are present in all other silk fibers, which makes the novelty of our work easier to see and easier to communicate.

3. Does the method introduced in this paper only work for this specific spider silk? Does the structure of this ribbon silk of the recluse spider differs greatly from most other spider silk? How much reference can the model introduced in this work provide for other kinds of spider silk?

#2.3 Response: As detailed in our #1.2 Response to a question of Reviewer #1, the method will not only be useful for other silks, but essentially for *any* proteinaceous material, for which our approach represents a significant progress regarding the assignment of peaks, the information about orientational order at the molecular level, and the understanding about which amino acids are a part of a particular secondary structure. For silks and spider silk in particular, the method will work very well, since all silks not only consist of proteins only, but also have very similar amino acid compositions and sequences.^{R10} As stated above, nanofibrils are present in other silks, and since we revealed the protein secondary structure of nanofibrils in unprecedented detail, our work is indeed a significant contribution to understanding the structure of all silks containing nanofibrils. For instance, by comparing our (purely nanofibrillar) spectra with spectra from silks with non-fibrillar components, it may be possible to draw conclusions about the makeup and protein secondary structures of their non-fibrillar phase(s).

A significant difference between the silk of the recluse spider and other spider silks is that most spider silk fibers feature cylindrical geometries, whereas the silk of the recluse spider ribbons feature a pseudo-2D geometry. As discussed in detail in our #1.4 Response to a question of Reviewer #1, this has implications for the calculations of the unpolarized spectra, which we had used for comparisons with published (unpolarized) spectra. This can be easily addressed by adjusting the formulas for the respective geometry being used. We highlighted this in the revised version of the Supplementary Information, where we provide the correct formulas for cylindrical geometry, in addition to the formulas for the geometry for the ribbon fiber. We have also expanded our discussion regarding the applicability of our approach for other spider silks and for other biomaterials in the revised manuscript.

4. What's the importance to provide the optical microscope pictures of the infrared and Raman experiments in Figure 1? maybe replaced with a picture of the spider silk ribbon under a polarizing microscope.

#2.4 Response: The reason why we provided the optical microscope pictures of the infrared and Raman samples in Figure 1 is that we can showcase the similarities in the sample preparation and geometry for these two experimental techniques. This approach ensures the datasets were collected under same conditions from these two techniques. Notably, Reviewer #3 explicitly asked whether Raman and FTIR spectra were taken from samples of the same size and geometry. Since other readers may have similar questions as they read the manuscript, we feel that it is best to leave the figure in place.

Furthermore, we took an image of a recluse ribbon under crossed polarizers, with the ribbon oriented at 45° relative to the axes of the polarizer and analyzer. However, we did not obtain sufficient contrast under

these conditions. This may be due to the fact that the thickness of the ribbon is only 40–50 nm, which may not be enough, given the expected, modest difference in refractive index between *X* and *Z* directions.

5. To provide a better understanding for a wider readership, a brief description is suggested to include: What is the difference between Raman spectroscopy and FTIR? Do these two characterization methods complement or verify the structural information?

#2.5 Response: We happily implemented the reviewer's suggestion and added the following passage to the introduction: "Raman spectroscopy, carried out with monochromatic visible light, probes polarizability of vibrations, while FTIR is sensitive to their associated dipole moments. For materials with inversion symmetry, Raman and IR modes are mutually exclusive, making the two techniques complementary and sensitive to the specimen's symmetry."^{R11,R12,R13,,}

Reviewer #3 (Remarks to the Author):

The manuscript deals with the critical issue of secondary structure determination in silk fibres. As rightly pointed by the authors, this is a topic highly researched and debated. The merit of the manuscript is in lifting the ambiguity often encountered during secondary structure assignments.

The work presented is exhaustive and well-grounded. The use of complementary methods and polarisation to enhance the information content.

Response: We thank the reviewer for the positive feedbacks.

A few general comments and questions

1. Silk fibres are often co-spun. Do the authors anticipate contribution from, for example, aciniform or piriform silk onto the significant ampullate silk under investigation?

#3.1 Response: We have worked with the recluse spider for over 10 years, and in all this time, there were only a couple of occasions on which we spotted silk that was not the ribbon silk that is the subject of this work, but small-diameter cylindrical silk, which was easily identifiable as something different. In other words, the production of other silk types can be neglected in the Chilean recluse spider. Moreover, the single ribbon samples used for this work were carefully examined with an optical microscope before the FTIR and Raman measurements. A representative high-magnification optical microscopy image of our Raman sample is shown as Figure 1e of the manuscript. The major ampullate (MA) ribbon silk can clearly be observed; any non-ribbon silk strands would have been spotted in such images. Therefore, we are sure that only MA silk was present in all samples used for this manuscript.

2. Vibrational spectroscopies (FTIR and Raman) are local probes. The direct consequence is that the size of the secondary structure may be unattainable. Therefore, the question is: Can the secondary structure's length cause some misinterpretation of the peak assignments?

#3.2 Response: We do not consider the size of the secondary structures to be an issue. We agree with the reviewer that the vibrational spectroscopies probe a particular molecular vibration and its local environment.^{R14} Determining the size of the secondary structures using our method would indeed be challenging. However, the size of the secondary structures, such as the crystalline β -sheets is already attainable using an entire range of techniques, including X-ray diffraction,^{R15} neutron scattering,^{S16} and

transmission electron microscopy;^{R15} it has already been established in the literature. It has been shown that the sizes of the secondary structures are largely dictated by the protein sequence^{R15} and hardly vary among many different species.^{R10} Therefore, since the size of the secondary structures is already known, determined by the sequence, and relatively invariant even among different species, the inability of our technique to determine their size does not pose a significant disadvantage.

In principle, however, the reviewer is correct. It has indeed been shown that for small secondary structures, such as groups of polypeptides with $N < 20$, the peak positions are somewhat sensitive to the size of the secondary structure. These frequency shifts are most likely caused by intra- and or intermolecular forces, such as transition dipole coupling.^{R17} As the number of polypeptide groups increases, their vibrational frequencies tend to a constant value.

3. Sample under investigation. It is unclear if the size of silk fibre/fibril under measurement is the same for the FTIR and Raman?

#3.3 Response: The size of the fiber/fibrils are the same for FTIR and Raman. We used the MA silk fibers collected from adult female *Loxosceles laeta* individuals to prepare both FTIR and Raman samples. Having worked with this particular silk for more than 10 years, we found such silk fibers are very uniform, featuring widths of 6–8 μm and thicknesses of 40–60 nm.¹⁸ As for the nanofibril dimension, we found these nanofibrils to feature an average width of 21.3 ± 3.3 nm and a typical thickness of 5–9 nm.^{R7} Over the years, we took hundreds of atomic force microscopy (AFM) images that fully confirmed this, without notable exceptions. Both the FTIR and Raman spectra presented in the manuscript we taken from individual silk ribbons, both featuring the described geometry and dimensions. Therefore, we have no doubt about a 1:1 correspondence between the sizes of the samples studied by FTIR and Raman.

Specific questions

Line 35: "that the outstanding mechanical properties of spider silk are already implemented at the level of an individual nanofibril". The statement is unclear. If referring to a tensile test, it will depend on the region considered. In a way, the purpose of the paper is to substantiate the statement.

#3.4 Response: We thank the reviewer for pointing this out. Reading our manuscript again, we realized that this statement in the introduction can be read as if it were a new claim or conclusion. However, it rather states an already established finding from one of our previous works.^{R7} In short, we had previously found that the nanofibrils within the silk ribbons adhere weakly to each other, relative to the strong tensile properties of the material. Based on the weak interaction between fibrils, we then concluded that the tensile properties of the entire ribbon are, essentially, the sum of the tensile forces supported by the individual fibrils. Knowing the number of nanofibrils within the ribbon ($\approx 2,500$), we were able to translate the breaking force of the ribbon (≈ 300 μN) into a breaking force of the individual nanofibril (≈ 120 nN). We changed the wording in the revised manuscript to make it clearer this is a reference to a previous publication rather than a new claim.

Line 237: "However, since α - and 3_1 -helices show opposite polarization preference, an alternate scenario leading to $\text{PZZ} \approx 0$ is a mixture of oriented α - and 3_1 -helices. Considering the amide-I peak alone, the two scenarios cannot be distinguished." The results are remarkable and highlight the strength of the

polarisation analysis. However, this assumes that the absorption cross-section of the two types of helices is similar, is that true? or that the two helices are of similar sizes and, therefore, contribution?

#3.5 Response: As we will explain in the following, the absorption cross-section is indeed *identical* between the two types of helices. The amide-I peak discussed here is due to a stretching vibration of the double bond between the oxygen atom and the carbon atom in the amino acid backbone, as shown in Figure 3a of the manuscript. The interaction cross section of this bond with infrared (FTIR) and visible (Raman) light only depends on the dipole moment (FTIR) and the polarizability (Raman) of this bond; it does not depend on the secondary structure of the protein (only the peak *position* does). Correspondingly, the heights of sub-peaks within the amide-I peak are indeed proportional to the number of amino acids. All our statements in the manuscript regarding the composition refer to volume fractions of the different secondary structures (and not *number* fractions of helices, etc.). Therefore, these statements only depend on the number of amino acids (which our method is probing), but not the *size* of the secondary structures. Furthermore, the dominant amino acids in spider silks have similar molecular weights.

References

- [R1] Eisoldt, L.; Smith, A.; Scheibel, T. Decoding the Secrets of Spider Silk. *Mater. Today* **14** (3), 80–86 (2011). [https://doi.org/10.1016/s1369-7021\(11\)70057-8](https://doi.org/10.1016/s1369-7021(11)70057-8)
- [R2] Su, I.; Buehler, M. J. “Nanomechanics of Silk: The Fundamentals of a Strong, Tough and Versatile Material” *Nanotechnology* **27** (30), 302001 (2016). <https://doi.org/10.1088/0957-4484/27/30/302001>
- [R3] J. Pérez-Rigueiro, M. Elices, G. R. Plaza, G. V. Guinea, “Similarities and Differences in the Supramolecular Organization of Silkworm and Spider Silk” *Macromolecules* **40**, 5360–5365 (2007). <http://pubs.acs.org/doi/abs/10.1021/ma070478o>
- [R4] Sponner, A.; Vater, W.; Monajembashi, S.; Unger, E.; Grosse, F.; Weisshart, K. “Composition and Hierarchical Organisation of a Spider Silk” *PLoS ONE* **2** (10), e998 (2007). <https://doi.org/10.1371/journal.pone.0000998>
- [R5] Augsten, K.; Mühlig, P.; Herrmann, C. “Glycoproteins and Skin-Core Structure in Nephila Clavipes Spider Silk Observed by Light and Electron Microscopy” *Scanning* **22** (1), 12–15 (2000). <https://doi.org/10.1002/sca.4950220103>
- [R6] S. Frisk, R. M. Ikeda, D. B. Chase, J. F. Rabolt, “Rotational Invariants for Polarized Raman Spectroscopy. Applied Spectroscopy” *Applied Spectroscopy* **57**, 1053 (2003). <https://doi.org/10.1366/00037020360695892>
- [R7] Q. Wang, H. C. Schniepp, “Strength of recluse spider’s silk originates from nanofibrils” *ACS Macro Letters* **7**, 1364–1370 (2018). <https://pubs.acs.org/doi/abs/10.1021/acsmacrolett.8b00678>
- [R8] D. I. Bower, “Investigation of molecular orientation distributions by polarized raman scattering and polarized fluorescence” *Journal of Polymer Science Part A-2: Polymer Physics* **10**, 2135–2153 (1972). <https://doi.org/10.1002/pol.1972.180101103>
- [R9] Q. Wang, H. C. Schniepp, “Nanofibrils as Building Blocks of Silk Fibers: Critical Review of the Experimental Evidence” *JOM* **71**, 1248–1263 (2019). <https://doi.org/10.1007/s11837-019-03340-y>
- [R10] Madurga et al., “Persistence and variation in microstructural design during the evolution of spider silk” *Scientific Reports* **5**, 14820 (2015). <https://doi.org/10.1038/srep14820>

- [R11] S. R. Elliot. *Physics of Amorphous Materials*, 2nd Ed. 1984.
- [R12] Byrne, H. J., Sockalingum, G. D., & Stone, N. (2010). Raman microscopy: complement or competitor. *Biomedical Applications of Synchrotron Infrared Microspectroscopy*, 11, 105-142; <https://doi.org/10.1039/9781849731997-00105>
- [R13] K. Hashimoto, V. R. Badarla, A. Kawai, T. Ideguchi, Complementary vibrational spectroscopy. *Nature Communications* 10 (2019). <https://doi.org/10.1038/s41467-019-12442-9>
- [R14] A. Barth, "Infrared spectroscopy of proteins" *Biochimica et Biophysica Acta* **1767**, 1073 (2007). <https://doi.org/10.1016/j.bbabi.2007.06.004>
- [R15] L. F. Drummy, B. L. Farmer, R. R. Naik, Correlation of the β -sheet crystal size in silk fibers with the protein amino acid sequence. *Soft Matter* 3, 877 (2007). <https://doi.org/10.1039/B701220A>
- [R16] D. Sapede, et al., "Nanofibrillar Structure and Molecular Mobility in Spider Dragline Silk" *Macromolecules* **38**, 8447–8453 (2005). <http://pubs.acs.org/doi/abs/10.1021/ma0507995>
- [R17] N. A. Nevskaya, Y. N. Chirgadze, "Infrared spectra and resonance interactions of amide-I and II vibrations of α -helix" *Biopolymers* **15**, 637–648 (1976). <https://doi.org/10.1002/bip.1976.360150404>
- [R18] H. C. Schniepp, S. R. Koebley, F. Vollrath, "Brown recluse spider's nanometer scale ribbons of stiff extensible silk" *Advanced Materials* **25**, 7028–7032 (2013). <http://dx.doi.org/10.1002/adma.201302740>

REVIEWERS' COMMENTS

Reviewer #1 (Remarks to the Author):

In accordance with my first reading, the corrected manuscript and the responses of the authors make this work highly recommendable for the silk domain, the spectroscopy community and more generally to people studying biomaterials. The manuscript has been improved accordingly to the issues and questions raised by the reviewers. Then, several original spectroscopy methods and procedures developed here may be useful to other scientists working on connected fields. I congratulate the authors for this nice piece of work which helps to advance the field and recommend this manuscript for publication.

I only have one short question: In the text, it is written that “Notably, both the 1671 cm⁻¹ Raman and 1633 cm⁻¹ IR peaks feature PZX values of ≈ -0.6 .” However, but maybe there is something I did not understand, the values in Table TS2, the Pzx value of the Raman 1617 component is +0.49. Is there an error/typo (the value is positive, while it should be negative)?

Reviewer #3 (Remarks to the Author):

The manuscript in its revised form lifts all the issues I had during the first round of reviews. The work is sound with far-reaching impact in the field of structure determination using vibrational spectroscopic. Within the silk field, the contribution highlights the need to truly understand the interplay between secondary structures in silks.

RESPONSES TO THE REVIEWER COMMENTS

The authors would like to thank all the reviewers of this manuscript for their thoughtful feedback and for the time they spent. Our detailed responses to all issues raised by the reviewers are provided in blue color below.

Reviewer #1 (Remarks to the Author):

In accordance with my first reading, the corrected manuscript and the responses of the authors make this work highly recommendable for the silk domain, the spectroscopy community and more generally to people studying biomaterials. The manuscript has been improved accordingly to the issues and questions raised by the reviewers. Then, several original spectroscopy methods and procedures developed here may be useful to other scientists working on connected fields. I congratulate the authors for this nice piece of work which helps to advance the field and recommend this manuscript for publication.

#1.1 Response: We thank the reviewer for the positive feedback.

I only have one short question: In the text, it is written that “Notably, both the 1671 cm^{-1} Raman and 1633 cm^{-1} IR peaks feature PZX values of ≈ -0.6 .” However, but maybe there is something I did not understand, the values in Table TS2, the Pzx value of the Raman 1617 component is +0.49. Is there an error/typo (the value is positive, while it should be negative)?

#1.2 Response: We are extremely grateful to the reviewer for this feedback. There was indeed an error for the P_{ZX} value of the Raman 1671 cm^{-1} peak in table Supporting Table 2, as pointed out by the reviewer. We have corrected the P_{ZX} value of the Raman 1671 cm^{-1} peak to -0.51 . When we did this update, we also noticed that a stale version of data was used to calculate the values in this table. Other values in this table were also updated with minor changes to reflect the latest data available. We are impressed by the reviewer’s great attention to detail, without which we would have most likely missed this error!

Reviewer #3 (Remarks to the Author):

The manuscript in its revised form lifts all the issues I had during the first round of reviews.

The work is sound with far-reaching impact in the field of structure determination using vibrational spectroscopic. Within the silk field, the contribution highlights the need to truly understand the interplay between secondary structures in silks.

#3.1 Response: We thank the reviewer for their positive feedback.